# Sobolev acceleration for neural networks

## Abstract

*Sobolev training*, which integrates target derivatives into the loss functions, has been shown to accelerate convergence and improve generalization compared to conventional $L^2$ training. However, the underlying mechanisms of this training method remain only partially understood. In this work, we present the first rigorous theoretical framework proving that Sobolev training accelerates the convergence of Rectified Linear Unit (ReLU) networks. Under a student–teacher framework with Gaussian inputs and shallow architectures, we derive exact formulas for population gradients and Hessians, and quantify the improvements in conditioning of the loss landscape and gradient-flow convergence rates. Extensive numerical experiments validate our theoretical findings and show that the benefits of Sobolev training extend to modern deep learning tasks.

## 1 Introduction

Deep learning has achieved remarkable success across numerous scientific and engineering domains, driven by advances in neural network architectures, such as U-Net (Ronneberger et al., 2015), ResNet (He et al., 2016), AlexNet (Krizhevsky et al., 2017), RNN encoder–decoder (Cho et al., 2014), and Transformer (Vaswani et al., 2017), and optimization methods like Adam (Kingma & Ba, 2014; Ruder, 2016) and RMSprop (Riedmiller & Braun, 1993). These innovations have led to major breakthroughs in computer vision (Shorten & Khoshgoftaar, 2019; Voulodimos et al., 2018) and natural language processing (Young et al., 2018; Otter et al., 2020). More recently, deep learning has gained traction in applied and computational mathematics, particularly scientific computing, where incorporating physical principles into training has enabled progress in modeling complex systems in fluid dynamics, materials science, and quantum mechanics (Karniadakis et al., 2021). In parallel, Neural Operators, such as Fourier Neural Operators (Li et al., 2020) and Deep Operator Networks (DeepONets) (Lu et al., 2021), have emerged as powerful architectures for learning mappings between infinite-dimensional function spaces, further expanding the applicability of deep learning to PDE-based systems.

To explain the success of neural networks, researchers have examined their expressive power. Building on Cybenko's foundational work on the universal approximation property of single-layer networks (Cybenko, 1989), later studies extended this to multilayer networks (Hornik et al., 1989) and to Sobolev spaces (Li, 1996), capturing not only function values but also their derivatives, an essential feature in many modern applications (Kissel & Diepold, 2020; Son et al., 2023; Vlassis & Sun, 2021). However, knowing that a neural network can approximate a function is of limited practical use unless we also understand how to find such a network. This highlights the importance of studying the (typically nonconvex) training dynamics of neural networks. While approximation theory is well developed, our understanding of optimization, especially under gradient-based methods, remains incomplete due to the complex landscape of neural loss functions. Nevertheless, recent work in the overparameterized regime, where the number of parameters exceeds the number of training samples, has revealed that gradient descent can efficiently reach global minima and exhibit near-linear convergence under certain conditions (Jacot et al., 2018; Du et al., 2018; Allen-Zhu et al., 2019; Du et al., 2019; Arora et al., 2019; Cocola & Hand, 2020).

*Sobolev training* was introduced by Czarnecki et al. (2017) as a framework for training neural networks by minimizing the Sobolev norm of the loss function, rather than relying solely on the traditional $L^2$ loss function. The authors demonstrated that Sobolev training can significantly reduce the sample complexity of training and yield substantially lower test error compared to the conventional $L^2$ loss function. Since then, Sobolev training has shown strong empirical performance across various scientific domains where derivative information is naturally available, such as network

compression, distillation, and physics-informed machine learning (Son et al., 2023; Vlassis & Sun, 2021; Mirzadeh et al., 2020; O'Leary-Roseberry et al., 2024). In cases where the derivative data is not directly accessible, researchers have explored incorporating numerical approximations, such as finite difference methods (Kissel & Diepold, 2020) and spectral differentiation techniques (Yu et al., 2023), to apply Sobolev training.

Researchers have found that minimizing the Sobolev norm instead of the $L^2$ norm can significantly accelerate the convergence of the $L^2$ error. This phenomenon, commonly referred to as *Sobolev acceleration*, has been observed in the original work of Czarnecki et al. (2017) as well as in Lu et al. (2022) for learning elliptic equations and in Son et al. (2023) for physics-informed neural networks (PINNs). However, to this date, there has not been a sound theoretical explanation of Sobolev acceleration. Existing analytical tools, particularly those addressing training with derivative-based losses (Cocola & Hand, 2020; Yu et al., 2023; Wang et al., 2022), fall short of explicitly explaining or quantifying the extent of this acceleration. The main goal of this study is to establish a theoretical foundation for understanding Sobolev acceleration.

Our key findings can be summarized at a high level as follows. The condition number of the Hessian of the objective function, the ratio of the maximum eigenvalue to the minimum eigenvalue, governs the convergence rate of many optimization algorithms (Nesterov et al., 2018; Shewchuk et al., 1994; Polyak, 1964; Beck & Tetruashvili, 2013). For instance, doubling the objective function doubles both extreme eigenvalues, leaving the condition number, and thus the convergence rate, unchanged. In contrast, *Sobolev training significantly increases the minimum eigenvalue of the Hessian, but barely increases the maximum eigenvalue, thereby improving the condition number of the objective*. In other words, matching the optimal gradients in addition to the function values gives new directions pointing toward the optimal parameter that are 'nearly uncorrelated' from those provided by the zeroth-order mismatch.

- Under Assumptions 2.1-2.3, we derive exact formulas for the Hessians of the $L^2$ and $H^1$ loss functions and prove that Sobolev training improves the condition number of the Hessian.
- We further provide a theoretical justification of Sobolev acceleration by analyzing the gradient flow dynamics, and quantify the acceleration for both $H^1$ and $H^2$ norms.
- We illustrate our analysis with numerical examples, demonstrating its generalization to practical scenarios, including neural network training under empirical risk minimization with stochastic gradient descent. We further validate the effect across various activation functions and architectures, such as Fourier feature networks (Tancik et al., 2020) and SIREN (Sitzmann et al., 2020).
- We also apply Sobolev training to modern deep learning tasks, including denoising autoencoders and diffusion models, and demonstrate both convergence acceleration and improved generalization ability.

## 1.1 RELATED WORKS

In Czarnecki et al. (2017), the authors provided evidence that Sobolev training reduces the sample complexity of training and achieves considerably higher accuracy and stronger generalization. Later, Lu et al. (2022) demonstrated implicit Sobolev acceleration, and Son et al. (2023) showed that Sobolev training expedited the training of neural networks for regression and PINNs. The impact of Sobolev training has extended across various fields, prompting extensive research. For instance, for PINNs, Son et al. (2023) introduced multiple loss functions tailored to Sobolev training, enhancing the training process. In another application, Vlassis & Sun (2021) harnessed Sobolev training to refine smoothed elastoplasticity models. Kissel & Diepold (2020) proposed to leverage approximated derivatives when the target derivatives are unavailable. The potential of Sobolev training was further exemplified by Cocola & Hand (2020), who demonstrated the global convergence of this approach for overparameterized networks. More recently, Yu et al. (2023) showcased how Sobolev loss functions could effectively manage the spectral bias of neural networks.

To analytically study the training dynamics of shallow neural networks with rectified linear unit (ReLU) activation under gradient descent, a line of research (Tian, 2017; Li & Yuan, 2017; Zhang et al., 2019) adopts the student–teacher framework, assuming the presence of a ground truth teacher network with the same architecture as that of the student network. Another line of research focuses on overparameterization, including Du et al. (2018); Chizat & Bach (2018); Arora et al. (2019); Allen-Zhu et al. (2019); Zou et al. (2020). Notably, Jacot et al. (2018) formulated the notion of the

neural tangent kernel (NTK), which is a constant kernel that characterizes the training of a neural network in the infinite width limit. Furthermore, Wang et al. (2022) extended this concept to PINNs. They derived the NTK for these networks and demonstrated its convergence to a constant kernel.

## 2 THEORETICAL RESULTS ON SOBOLEV ACCELERATION

### 2.1 PROBLEM SETUP

Sobolev training for fitting a neural network $g(x; w)$ to $f$ in the expected loss minimization with data distribution $\mathcal{P}$ on $\mathbb{R}^d$ can be formulated as

$$\min_{w \in \mathbb{R}^d} \left[ \mathcal{H}(w) := \mathbb{E}_{x \sim \mathcal{P}} \left[ \frac{1}{2}(g(x; w) - f(x))^2 + \frac{1}{2} \|\nabla_x g(x; w) - \nabla_x f(x)\|_2^2 \right] \right], \quad (1)$$

which is to minimize the expected $H^1$-distance between $g(x; w)$ and $f$. Omitting the first-order term from above, we get the usual $L^2$-training. One can add higher-order mismatches. For instance, the $H^2$-training would mean adding the Hessian mismatch term $\|\nabla_x^2 g(x; w) - \nabla_x^2 f(x)\|_F^2$. Throughout this paper, $\|\cdot\|$ denotes the standard $L^2$ norm for vectors and the Frobenius norm for matrices, unless otherwise specified.

To facilitate the theoretical analysis, we adopt the standard student–teacher assumption for the model class $\{g(x; w)\}_w$, commonly used in the literature (Tian, 2017; Goldt et al., 2019; Akiyama & Suzuki, 2021):

**Assumption 2.1** (Student-teacher setting). There exists an unknown *teacher parameter* $w^*$ for which $f(\cdot) = g(\cdot; w^*)$.

This assumption provides an explicit relationship between the target function and its derivative. Proving the acceleration of Sobolev training becomes challenging without the assumption, owing to the absence of relational information between the target function and its derivative. For instance, Cocola & Hand (2020) showed the convergence of Sobolev training for neural networks in the NTK regime. However, since the labels for the target and its derivative were defined as separate vectors, this approach could not provide insights into the relationship between the two components in the Sobolev loss function, thereby hindering further derivation of acceleration results.

We focus on a class of two-layer ReLU networks with unit weights in the second layer. This model has been considered in the literature on two-layer ReLU networks (Tian, 2017; Li & Yuan, 2017; Cocola & Hand, 2020; Akiyama & Suzuki, 2023).

**Assumption 2.2** (Two-layer ReLU network). $g(x; w) = \sum_{j=1}^K \sigma(w_j^\top x)$ where $w = [w_1, \ldots, w_K] \in \mathbb{R}^{d \times K}$ and $\sigma(t) = \max(0, t)$ with $K \geq 1$ ReLU nodes in the hidden layer. In the special case $K = 1$, $w$ reduces to a vector in $\mathbb{R}^d$.

Lastly, following Tian (2017); Li & Yuan (2017); Brutzkus & Globerson (2017); Wu et al. (2019), we consider a standard Gaussian data distribution. This allows us to derive analytical expressions of the population gradients and Hessians.

**Assumption 2.3** (Gaussian population). The data distribution $\mathcal{P}$ is the standard Gaussian $N(0, I_{d \times d})$.

**Remark 2.4.** The Gaussian population assumption is used solely to obtain closed-form expressions for the population gradients and Hessians. In principle, the same framework applies to any input distribution for which these expectations can be computed in closed form (e.g., uniform or other analytically tractable distributions).

Based on Assumptions 2.1-2.3, the $H^1$ population loss function in equation 1 specializes as

$$\mathcal{L}(w) := \mathbb{E}_{N(0, I_{d \times d})} \left( \frac{1}{2N} \sum_{j=1}^N (g(x_j; w) - g(x_j; w^*))^2 \right),$$

$$\mathcal{J}(w) := \mathbb{E}_{N(0, I_{d \times d})} \left( \frac{1}{2N} \sum_{j=1}^N \|\nabla_x g(x_j; w) - \nabla_x g(x_j; w^*)\|^2 \right), \quad \mathcal{H}(w) = \mathcal{L}(w) + \mathcal{J}(w).$$

**Remark 2.5.** Under assumptions 2.1 and 2.3, we rigorously prove for linear models that Sobolev training yields both faster convergence and improved generalization in Appendix A.

## 2.2 EXACT OPTIMIZATION LANDSCAPE FOR A SINGLE RELU NODE ($K = 1$) AND GRADIENT DESCENT

Our first result gives the exact optimization landscape for the $L^2$ and the $H^1$ training in the case of a single ReLU node ($K = 1$). Namely, we obtain exact analytic formulas for the population Hessian of the loss functions $\mathcal{L}$ and $\mathcal{H}$. In particular, this yields analytical expressions for the condition numbers of these loss functions. Recall that for a real symmetric matrix $A$, let $\kappa(A) = \frac{\lambda_{\max}(A)}{\lambda_{\min}(A)}$ denote its condition number, where $\lambda_{\max}(A)$ and $\lambda_{\min}(A)$ denote the maximum and the minimum modulus eigenvalues of $A$, respectively.

**Theorem 2.6.** *[Comparison of the optimization landscapes] Assume 2.1, 2.2, 2.3 hold with $K = 1$. Let $\theta$ denotes the angle between $w$ and $w^*$. Then we have*

$$\nabla_w^2 \mathcal{L} = \frac{1}{2}I - \alpha(uu^\top - \cos(\theta)vu^\top + \sin^2(\theta)I)(I - vv^\top),$$

$$\nabla_w^2 \mathcal{H} = I - \alpha(2uu^\top - \cos(\theta)vu^\top + \sin^2(\theta)I)(I - vv^\top) \tag{2}$$

*where $\alpha = \frac{\|w^*\|}{2\pi\|w\|\sin(\theta)}$, $u = \frac{w^*}{\|w^*\|}$, and $v = \frac{w}{\|w\|}$. Furthermore, if $\frac{\|w^*\|\sin(\theta)}{\|w\|} < \frac{\pi}{2}$, then*

$$\kappa(\nabla_w^2 \mathcal{H}) = \frac{1}{1 - 4\alpha\sin^2(\theta)} < \frac{1}{1 - 3\alpha\sin^2(\theta)} = \kappa(\nabla_w^2 \mathcal{L}).$$

Our result above gives an explicit impact of the $H^1$ Sobolev training on the optimization landscape. Even in the single ReLU node setting, we observe that such an impact on the Hessian is nonlinear.

**Remark 2.7** ($\mathcal{L}$ vs. $2\mathcal{L}$ vs. $\mathcal{L} + \mathcal{J}$). Minimizing $\mathcal{L}$ or $2\mathcal{L}$ yields the same convergence rate, as both have identical condition numbers. In contrast, as noted in Theorem 2.6, Sobolev training (via $\mathcal{L} + \mathcal{J}$) improves the condition number of the Hessian, thereby accelerating convergence.

**Remark 2.8** (Larger basin of attraction for Sobolev training). Since $\lambda_{min}(\nabla_w^2 \mathcal{L}) > 0 \Leftrightarrow \sin(\theta) < \frac{\pi\|w\|}{2\|w^*\|}$ and $\lambda_{min}(\nabla_w^2 \mathcal{H}) > 0 \Leftrightarrow \sin(\theta) < \frac{2\pi\|w\|}{3\|w^*\|}$, $\mathcal{H}$ is strictly convex over the larger region $S' = \{w : \sin(\theta) < \frac{2\pi\|w\|}{3\|w^*\|}\}$ whereas $\mathcal{L}$ is strictly convex on the smaller region $S = \{w : \sin(\theta) < \frac{\pi\|w\|}{2\|w^*\|}\}$. Consequently, while $\mathcal{L}$ may attain saddle points and spurious local minima in $S' \setminus S$, $\mathcal{H}$ is free from them, all without compromising the condition number.

Many standard optimization algorithms, such as Gradient Descent (GD) (Nesterov et al., 2018), Conjugate Gradient (CG) (Shewchuk et al., 1994), Momentum and Nesterov's Accelerated Gradient (NAG) (Polyak, 1964), and Coordinate Descent (CD) (Beck & Tetruashvili, 2013), converge fast for problems that have a small condition number. In the following corollary of Theorem 2.6, we show that one-step GD update with a common and sufficiently small stepsize decreases the parameter estimation error faster under $H^1$ and under $L^2$ training.

**Corollary 2.9.** *[Single-GD-step improvement] Assume 2.1, 2.2, 2.3 hold with $K = 1$. Consider the two gradient descent updates*

$$w_{new}^{(1)} = w_{old} - \eta\nabla_w\mathcal{L}(w_{old}), \quad w_{new}^{(2)} = w_{old} - \eta\nabla_w\mathcal{H}(w_{old})$$

*with stepsize $\eta > 0$. Then there exists explicit functions $C = C(w_{old}, w^*) > 0$ and an explicit function $F = F(w_{old}, w^*; \eta)$ such that whenever $\eta \leq C$, $F > 0$ and*

$$\|w_{new}^{(2)} - w^*\| \leq \|w_{new}^{(1)} - w^*\| - F.$$

## 2.3 SOBOLEV ACCELERATION ON GRADIENT FLOW

Next, we compare the dynamics of the squared error for the parameter estimation $V(w) = \|w - w^*\|_2^2$ under the gradient flows of the loss functions defined by different Sobolev norms, $L^2$, $H^1$ (up to the first derivative), and $H^2$ (up to the second derivative) for general $K \geq 1$ ReLU nodes. We will derive analytical formulas of the dynamics under the gradient flow $\dot{w} = -\nabla_w\mathbb{E}_{x\sim N(0,I)}(J(x; w))$ of the population loss function $\mathbb{E}_{x\sim N(0,I)}(J(x; w))$, where $J$ denotes the corresponding per-sample loss function.

Gradient flows are often more convenient to analyze than gradient descent, as they avoid issues related to overly large step sizes. In particular, a faster convergence rate for gradient flow typically

reflects a larger minimum eigenvalue of the Hessian (Boyd & Vandenberghe, 2004). However, we note that this alone does not imply an improved condition number; a corresponding analysis of the maximum eigenvalue, similar to that in Theorem 2.6, is also necessary. Analyzing the maximum eigenvalue, however, appears to be more challenging in the general setting. We hope that our gradient flow analysis offers insight into this intriguing problem.

### 2.3.1 $H^1$-FLOW ACCELERATION FOR A SINGLE ReLU NODE ($K = 1$)

We first analyze gradient flow under training with the same setting of $K = 1$ ReLU node. We begin by recalling a result of Tian (2017) for the gradient flow for $L^2$-training with single ReLU node:

**Theorem 2.10** (Theorem 5 in Tian (2017)). *Assume 2.1, 2.2, 2.3 hold with $K = 1$. Consider the gradient flow $\dot{w} = -\nabla_w \mathcal{L}(w)$, where $\mathcal{L}$ is given in equation 2, and $V(w) = \|w - w^*\|_2^2$. Suppose that an initial parameter $w^0$ satisfies $\|w^0 - w^*\| < \|w^*\|$, then $\frac{dV}{dt} = -(w - w^*)^\top \nabla_w \mathcal{L} < 0$ and $w^t \to w^*$ as $t \to \infty$.*

This theorem states that for a neural network with a single ReLU node, global convergence can be achieved depending on the initial parameter $w^0$. In the next theorem, we show that by using the $H^1$ loss function, the decay of $V$ can be accelerated.

**Theorem 2.11.** *Assume 2.1, 2.2, 2.3 hold with $K = 1$. Consider the gradient flow $\dot{w} = -\nabla_w \mathcal{H}(w)$, where $\mathcal{H}$ is given in equation 2, and $V(w) = \|w - w^*\|_2^2$. Suppose that $\|w^0 - w^*\| < \|w^*\|$. Then,*

$$\frac{dV}{dt} = -(w - w^*)^\top \nabla_w \mathcal{H} \le -(w - w^*)^\top \nabla_w \mathcal{L} - \lambda(\theta)(\|w\|^2 + \|w^*\|^2) < 0,$$

*where $\lambda(\theta) = (2\pi - \theta) - \sqrt{\theta^2 + (2\pi - \theta)^2 \cos^2 \theta} \ge 0$ with $\theta$ denoting the angle between $w$ and $w^*$. Therefore, the decay of $V$ is accelerated by using the Sobolev loss function. Moreover, since the rate of acceleration $\lambda(\theta)$ is an increasing function of $\theta \in [0, \pi/2)$, the strength of acceleration increases as $\theta$ increases.*

**Remark 2.12.** Here, $\lambda(\theta)$ serves only as a quantitative lower bound on this effect. The acceleration never vanishes when $\theta = 0$ (see the proof).

### 2.3.2 $H^2$-FLOW ACCELERATION FOR A SINGLE ReLU$^2$ NODE ($K = 1$)

We now demonstrate the same effect for higher-order derivatives. As the ReLU function is now twice weakly differentiable, we consider a neural network with a single ReLU-square node, $g(x) = \left(\sigma(w^\top x)\right)^2$, where $w, x \in \mathbb{R}^d$, which has been widely considered in the literature (Yu et al., 2018; Cai & Xu, 2019). We show the global convergence of the neural network with one ReLU$^2$ node in $L^2$ and the convergence acceleration in $H^1$, $H^2$ spaces.

**Theorem 2.13.** *Assume 2.1 and 2.3 hold. Suppose $g(x; w) = \left(\sigma(w^\top x)\right)^2$, a two-layer network with a single ReLU$^2$ node. Consider the gradient flow $\dot{w} = -\nabla_w \mathcal{I}(w)$, where $\mathcal{I}(w) = \mathcal{I}_1(w) + \mathcal{I}_2(w) + \mathcal{I}_3(w)$ is the $H^2$ population loss with $\mathcal{I}_1 = \mathcal{L}$, $\mathcal{I}_2 = \mathcal{J}$, and $\mathcal{I}_3 = \mathbb{E}\left(\|\nabla_x^2 g(x; w) - \nabla_x^2 g(x; w^*)\|^2\right)$. If $\|w^0 - w^*\| < \|w^*\|$ then,*

$$-(w - w^*)^\top \nabla_w \mathcal{I}_j(w) < 0, \text{ for } j = 1, 2, 3,$$

*and hence, the decay of $V = \|w - w^*\|^2$ is accelerated under the gradient flow minimizing the higher order Sobolev loss functions.*

### 2.3.3 $H^1$-FLOW ACCELERATION FOR A TWO-LAYER GENERAL ReLU NETWORK ($K > 1$)

In this section, we consider the general ReLU network in Assumption 2.2 with $K \ge 1$ hidden nodes. Following Tian (2017), we focus on a special case that the teacher parameters $\{w_j^*\}_{j=1}^K$ form an orthonormal basis, where $w_j^* = P_j w^*$, for an orthogonal matrix $P_j$, and $\{P_j\}_{j=1}^K$ forms a cyclic group in which $P_j$ circularly shift dimension. Under this setting, we are now ready to show $H^1$ acceleration for a two-layer ReLU network by comparing the gradient flows $\dot{w} = -\nabla_w \mathcal{L}(W)$, and $\dot{w} = -\nabla_w \mathcal{H}(W)$. The setting of the following result is similar to (Tian, 2017, Theorem 7) for $L^2$-training.

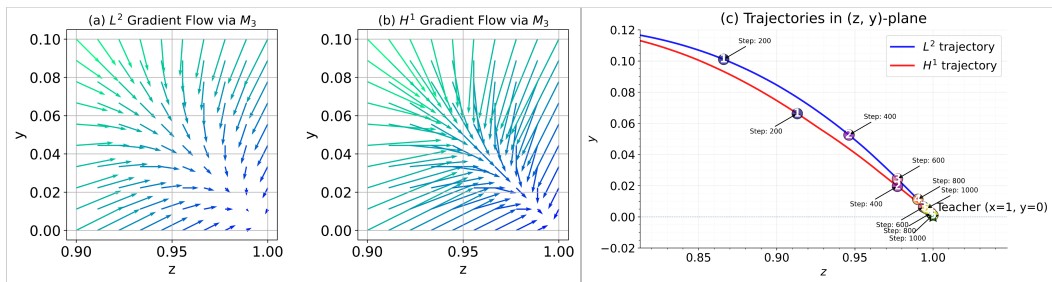

Figure 1: (a), (b) Vector fields induced by the $L^2$ and $H^1$ gradient flows, respectively, illustrating their respective convergence behaviors near the target $(z, y) = (1, 0)$. (c) Optimization trajectories, where the $L^2$ path is shown in blue and the $H^1$ path in red. Markers with matching colors and labels are placed every 200 steps to indicate the progression along each trajectory, and the green star denotes the target point $(z, y) = (1, 0)$.

**Theorem 2.14.** *Assume 2.1, 2.2, 2.3 hold with $K \geq 1$. If the teacher parameters $\{w_j^*\}_{j=1}^K$ form an orthonormal basis and a student parameter $W$ is initialized to be*

$$w_l = yw_1^* + \cdots + yw_{l-1}^* + zw_l^* + yw_{l+1}^* + \cdots + yw_K^*, \qquad (3)$$

*under the basis of $\{w_j^*\}_{j=1}^K$, where $(z, y) \in \Omega = \{z \in (0, 1], y \in [0, 1], z > y\}$, then the following holds.*

*(1) The student parameter $w_l$ converges to $w_l^*$ under the gradient flow $\dot{w}_l = -\nabla_{w_l}\mathcal{H}(W)$, i.e., $(z, y)$ converges to $(1, 0)$.*

*(2) Near $(z, y) = (1, 0)$, the gradient flows can be linearized to 2-d dynamical systems:*

$$L^2 \text{ gradient flow} : \begin{pmatrix} \dot{z} \\ \dot{y} \end{pmatrix}_{L^2} \approx -M_3 \begin{pmatrix} z - 1 \\ y \end{pmatrix},$$

$$H^1 \text{ gradient flow} : \begin{pmatrix} \dot{z} \\ \dot{y} \end{pmatrix}_{H^1} \approx -2M_3 \begin{pmatrix} z - 1 \\ y \end{pmatrix},$$

*and the eigenvalues of $M_3$ are $\lambda_1(M_3) = \frac{\pi}{2}$, and $\lambda_2(M_3) = \frac{\pi}{2}(K + 1)$. Hence, $M_3$ is positive definite for all K, and the convergence is accelerated. (see Figure 1.)*

*(3) When $z, y$ are initialized such that $z = y \in (0, 1]$, the $L^2$ gradient flow converges to the saddle point $z = y = z_{L^2}^* = \frac{1}{\pi K}(\sqrt{K-1} - \arccos(\frac{1}{\sqrt{K}}) + \pi)$, and the $H^1$ gradient flow converges to the saddle point $z = y = z_{H^1}^* = \frac{1}{2\pi K}(\sqrt{K-1} + 2\pi - 2\arccos(\frac{1}{\sqrt{K}}))$ and $z_{L^2}(t) = (z(0) - z_{L^2}^*)e^{-K/2t}$ and $z_{H^1}(t) = (z(0) - z_{H^1}^*)e^{-Kt}$. Hence, the convergence is accelerated.*

Lastly, we extend the result in Theorem 2.14 by allowing a more general initialization than the one in (3).

**Theorem 2.15.** *If the teacher parameters $\{w_j^*\}_{j=1}^K$ form an orthonormal basis and the student parameters are initialized as the symmetric Toeplitz matrix:*

$$\begin{pmatrix} w_1 \\ w_2 \\ w_3 \\ \vdots \\ w_K \end{pmatrix} = \begin{pmatrix} t_1 & t_2 & t_3 & \cdots & t_K \\ t_2 & t_1 & t_2 & \cdots & t_{K-1} \\ t_3 & t_2 & t_1 & \cdots & t_{K-2} \\ \vdots & \vdots & \vdots & \ddots & \vdots \\ t_K & t_{K-1} & t_{K-2} & \cdots & t_1 \end{pmatrix} \begin{pmatrix} w_1^* \\ w_2^* \\ w_3^* \\ \vdots \\ w_K^* \end{pmatrix},$$

*then under the linearized $L^2$ and $H^1$ gradient flows, $\mathbb{T}_{L^2} = (t_1 - 1, t_2, \ldots, t_K)$ and $\mathbb{T}_{H^1} = (t_1 - 1, t_2, \ldots, t_K)$ follow*

$$\dot{\mathbb{T}}_{L^2} = -M\mathbb{T}_{L^2}, \quad \dot{\mathbb{T}}_{H^1} = -2M\mathbb{T}_{H^1}$$

*for a positive definite matrix $M$.*

**Remark 2.16.** Theorem 2.15 analyzes the linearization of the gradient flow around the ground-truth parameters. Therefore, the convergence acceleration describes the local behavior of the dynamics in a neighborhood of $(t_1, t_2, \ldots, t_K) = (1, 0, \ldots, 0)$

# 3 EXPERIMENTS

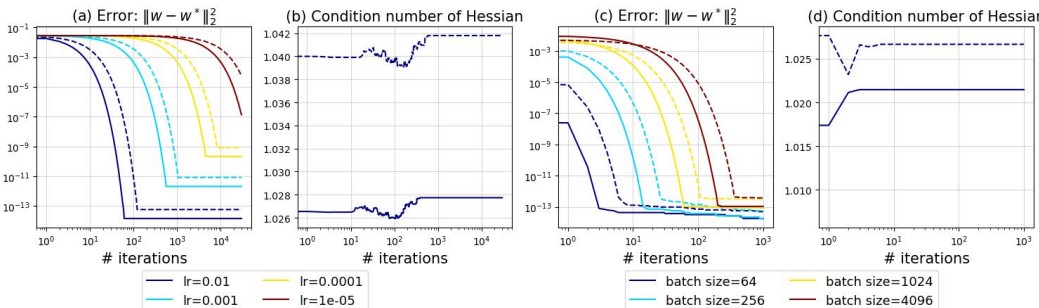

Figure 2: Comparison of the convergence behavior of $L^2$ training (dashed lines) and $H^1$ training (solid lines). (a) Errors for different learning rates. (b) Condition number of the Hessian during training with learning rate $0.01$. (c) Errors for different batch sizes. (d) Condition number of the Hessian during training with batch size $64$.

We numerically validate Sobolev acceleration across a range of tasks, aiming to relax the restrictive assumptions made in our theoretical analysis progressively. While our results were derived under the population loss with Gaussian inputs, two-layer ReLU networks, and the student–teacher setting, practical deep learning involves empirical loss minimization via stochastic optimization, arbitrary data distributions, complex network architectures, and general target functions. Our experiments investigate whether Sobolev acceleration persists once the idealized assumptions are lifted, providing evidence that the phenomenon extends beyond the narrow theoretical regime and manifests robustly in realistic deep learning tasks. All training and inference are conducted using a single NVIDIA A6000 GPU.

## 3.1 EMPIRICAL RISK MINIMIZATION WITH SGD

We randomly generate $w, w^* \in \mathbb{R}^d$ such that $\|w - w^*\| < \|w^*\|$. We use SGD to minimize the empirical loss functions $\frac{1}{2N} \sum_{j=1}^{N} (g(x_j; w) - g(x_j; w^*))^2$ and $\frac{1}{2N} \sum_{j=1}^{N} (g(x_j; w) - g(x_j; w^*))^2 + \|\nabla_x g(x_j; w) - \nabla_x g(x_j; w^*)\|^2$, where $g(x; w) = \sigma(w^\top x)$ and N=10,000. We explore a range of relatively large learning rates: $[1e - 1, 1e - 2, 1e - 3, 1e - 4]$. Panels (a) and (c) of Figure 2 illustrate the errors $\|w - w^*\|^2$ during training for various learning rates and batch sizes in log–log scales, respectively. Panels (b) and (d) present the condition number of Hessian during training for learning rate $0.01$ and batch size $64$, respectively. Since the trajectory of the condition number exhibits little variation across different configurations, we omit the corresponding plots from the figure. As shown, Sobolev training accelerates convergence and leads to better local minima and improves the Hessian conditioning.

## 3.2 SOBOLEV ACCELERATION FOR VARIOUS ARCHITECTURES

We first illustrate Sobolev acceleration on fully connected networks with different activations. The target is $f(x, y) = \sin(10(x + y)) + (x - y)^2 - 1.5x + 2.5y + 1$ on $(x, y) \in [1, 4] \times [-3, 4]$. A 2-64-64-64-1 network is trained with Adam ($1e - 4$) for 50,000 epochs, repeated 50 times with independent initializations. We compare ReLU, Leaky ReLU, GeLU, Tanh, and Sine. Figure 3 shows that Sobolev losses ($H^1, H^2$) accelerate convergence for all activations, with the strongest effect for Sine, which is particularly effective in capturing high-frequency features (Sitzmann et al., 2020; Yu et al., 2023).

Neural networks exhibit spectral bias toward low frequencies (Rahaman et al., 2019). Fourier features alleviate this limitation (Tancik et al., 2020), while SIRENs (Sitzmann et al., 2020), which use

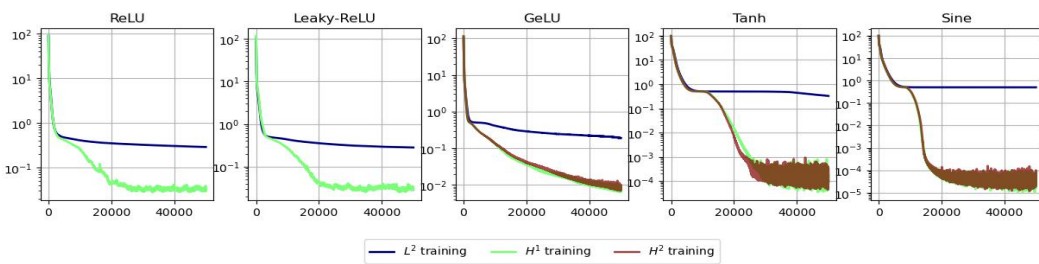

Figure 3: Test error versus training epochs for various activation functions. In all cases, Sobolev training leads to faster convergence and improved test error.

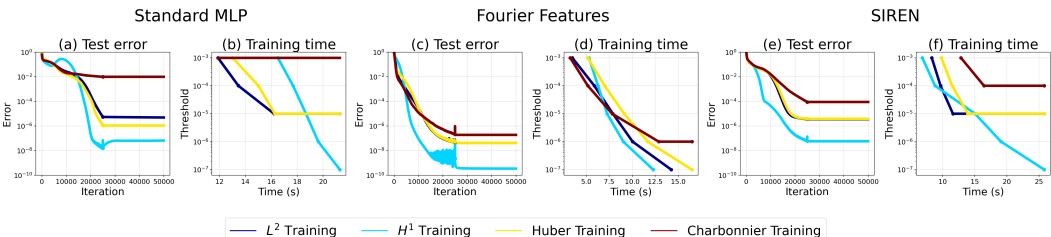

Figure 4: Comparison of test errors and training times across different architectures trained with $L^2$, $H^1$, Huber, and Charbonnier losses.

periodic activations with principled initialization, are also well suited for multi-scale representation. We show that integrating Sobolev training with these architectures further improves robustness. For the multi-scale target $f(x) = x + \sin(2\pi x^4)$ on $[-1, 1]$ (Wang et al., 2021), we train a 64-64-1 Fourier feature network (64 random features) and a 1-64-64-64-1 SIREN. Using Adam (1e−5) for 25,000 epochs followed by L–BFGS (1e−2) for an additional 25,000 epochs (Rathore et al., 2024), we adopt this training schedule to ensure stable optimization and to clearly expose the networks' ability to alleviate the high-frequency oscillations.

As shown in Figure 4 (a), (c), and (e), the $H^1$ loss yields the lowest approximation error across all architectures and loss functions. Figure 4 (b), (d), and (f) further report the wall-clock time required to reach specified error thresholds for the standard MLP, Fourier feature networks, and SIREN. Although $H^1$ training can be slower than the other losses at very early stages, it reaches lower error levels in significantly less time as training progresses. The panels clearly show that $H^1$ training attains the same accuracy in substantially less time than $L^2$, Huber, and Charbonnier training, thereby highlighting the Sobolev acceleration.

### 3.3 SOBOLEV TRAINING FOR THE DENOISING AUTOENCODERS

Autoencoders can be employed for image denoising by training the network to map noisy input images to their corresponding clean versions. This task can be naturally integrated with Sobolev training, as first considered in Yu et al. (2023). We present several numerical experiments demonstrating the accelerated convergence and improved generalization ability achieved through Sobolev training using the denoising autoencoders equipped with Convolutional Neural Networks (CNNs). We adopt the numerical differentiation technique from Yu et al. (2023) to implement the $H^1$ loss function. A comparison of different numerical approximations is provided in Appendix C.2.

We utilize a simple autoencoder comprising an encoder and a decoder, each consisting of three convolution layers with LeakyReLU activations. The Adam optimizer with a learning rate of $5e − 3$ is employed. The input image is contaminated with two types of additive noise: a Gaussian noise $\epsilon_1$ and a deterministic noise with a specific amplitude and frequency $\epsilon_2$. During training, the models take noisy images, generated by adding $\epsilon_1 \sim N(0, 1/4)$ and $\epsilon_2 = 0.3\sin(2\pi(x + y))$ to the clean images, as inputs, and are trained to output clean ones. Subsequently, the trained autoencoders are tested with significantly amplified noise levels: $\epsilon_1 \sim N(0, 1)$ and $\epsilon_2 = 0.3\sin(20\pi(x + y))$. The testing phase aims to assess the improved generalization performance of Sobolev training.

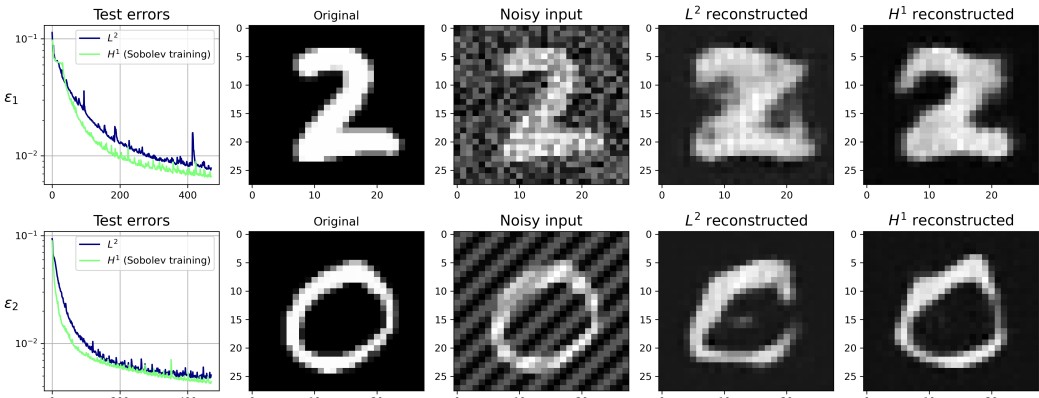

Figure 5: Results of the denoising autoencoders for the $\epsilon_1$ noise case are presented in the top row, and those for the $\epsilon_2$ noise case are shown in the bottom row. The first column illustrates convergence acceleration, the second and third columns display clean and noisy inputs, respectively. The fourth and fifth columns present the $L^2$ and $H^1$ reconstruction results.

The first column of Figure 5 illustrates the convergence acceleration achieved through Sobolev training in both noise settings. The second and third columns show the clean and noisy inputs, respectively. The third and fourth columns present the test reconstruction results of $L^2$ and $H^1$ trained autoencoders, respectively. These results highlight the enhanced generalization ability achieved through Sobolev training. We provide further experimental results for the autoencoder task in Appendix C.3.

## 3.4 SOBOLEV TRAINING FOR THE DIFFUSION MODELS

The diffusion model has achieved remarkable success in image synthesis and various generative tasks (Croitoru et al., 2023). A representative example is the Denoising Diffusion Probabilistic Model (DDPM) (Ho et al., 2020), which consists of two main processes: a forward process that progressively corrupts the data by adding Gaussian noise until it becomes pure noise ($\mathcal{N}(\mathbf{0}, I)$), and a reverse process that aims to reconstruct the original data by iteratively denoising the corrupted samples. In the reverse process of DDPM, the network takes a noisy image as input and predicts the original clean image. As a result, training a DDPM reduces to learning a denoising autoencoder with a certain variance schedule (Ho et al., 2020). Therefore, Sobolev training can naturally be applied to the diffusion model. In this experiment, we apply Sobolev training to demonstrate accelerated convergence toward a better generative model. We trained our diffusion model for 100 epochs with the ADAM optimizer with a learning rate of $1e - 4$. We use the CelebA-HQ dataset (Karras et al., 2017) with a batch size of 48 after downsampling the data into $128 \times 128$ resolution. The diffusion model employs a U-Net architecture with 1,000 diffusion timesteps during training and 20 sampling steps using the Denoising Diffusion Implicit Model (DDIM) (Song et al., 2020) during inference.

Table 1: Comparison of per-step computational cost between the $L^2$ and $H^1$ losses. Although the $H^1$ loss requires more time to compute than the $L^2$ loss, its contribution to the overall step time remains small (1–3%), resulting in only a minor difference in the actual wall-clock training time.

| Loss | Steps/Epoch | Loss time (s) | Step time (s) | Loss/Step (%) |
|------|-------------|---------------|---------------|---------------|
| $L^2$ Loss | 938 | 0.01234 | 0.98607 | 1.3 |
| $H^1$ Loss | | 0.02871 | 1.00283 | 2.9 |

The left panel of Figure 6 presents the Fréchet Inception Distance (FID) scores for both training methods, showing that Sobolev training leads to faster convergence of the FID score. The center and right panels display sample images generated using the models trained with $L^2$ and $H^1$ loss functions, respectively. Notably, the model trained with the $H^1$ loss generates images that appear more realistic and closely resemble human faces. Regarding computational cost, Table 1 shows that computing $H^1$

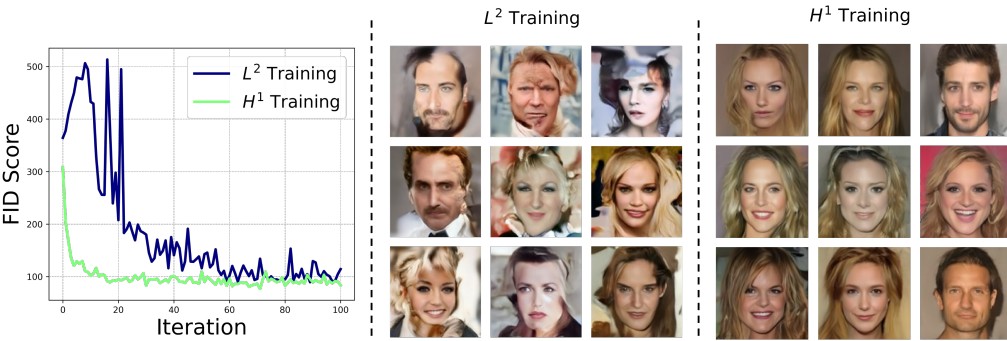

Figure 6: Results of the diffusion models on the CelebA-HQ dataset. The left panel presents the FID scores for each training method: $L^2$ training (blue) and $H^1$ training (green). The center panel presents samples generated by the $L^2$ trained model, while the right panel shows samples from the $H^1$ trained model.

loss is about $2.3\times$ slower than $L^2$ loss when the loss computation is measured in isolation. However, a single training step takes roughly one second, and the loss evaluation accounts for only 1–3% of that time. Thus, the additional cost introduced by the $H^1$ loss has only a minor effect on the actual wall-clock time per step. Given that each epoch contains 938 steps, the total difference per epoch amounts to only a few seconds. A more detailed analysis, including algorithmic complexity, GFLOPs, and peak memory usage, is provided in Appendix D. These results demonstrate the effectiveness of Sobolev training and suggest its potential for broader application in modern deep learning tasks.

## 4 CONCLUSION

Sobolev acceleration is a convergence acceleration phenomenon in neural network training that has been consistently reported in the literature. This paper provides the first rigorous theoretical foundation of Sobolev acceleration by analyzing the Hessians of the loss landscapes and the gradient flow dynamics in the student–teacher setting for shallow ReLU networks. Beyond theoretical findings, we further present several empirical observations showing that Sobolev acceleration is a general phenomenon across modern deep learning tasks, such as diffusion models, where it yields both faster convergence and improved generalization.

As a concluding remark, we emphasize the importance of further developing the theoretical foundation of this work. In particular, extending the gradient dynamics analysis beyond the idealized setting of Gaussian inputs and shallow architecture to deeper and more complex architectures remains a central open challenge. Addressing this challenge will help bridge the gap between rigorous mathematical theory and practical deep learning, ultimately providing both a deeper understanding and broader applicability.

## REPRODUCIBILITY STATEMENT

We have taken several steps to ensure the reproducibility of our results. All theoretical assumptions required for the analysis are explicitly stated in Section 2, and complete proofs of the main theorems are provided in Appendix B. For the empirical studies in Section 3, we release the full source code and scripts as supplementary materials, enabling others to reproduce the experiments across different architectures and tasks. Detailed descriptions of model architectures, training procedures, and hyperparameters are included in Section 3 and Appendix C. Together, these efforts provide a comprehensive basis for reproducing both the theoretical and experimental results presented in this work.

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

## A   SOBOLEV TRAINING FOR LINEAR MODELS

As an illustrative example, we provide a proposition, demonstrating that Sobolev training accelerates the gradient descent and improves the generalization error of the linear model.

**Proposition A.1.** *Let $X \in \mathbb{R}^{N \times d} \sim P_{data}$ denote the given data matrix and $y = Xw^* + \epsilon \in \mathbb{R}^N$ be corresponding labels, where $\epsilon \sim \mathcal{N}(0, \sigma^2 I)$. Consider the linear model $g(x; w) = w^T x$. Define the following loss functions:*

$$\mathcal{L}(w) = \frac{1}{2} \sum_{i=1}^{N} (w^T x_i - w^{*T} x_i)^2 = \frac{1}{2} \|Xw - Xw^*\|^2,$$

$$\mathcal{H}(w) = \frac{1}{2} \sum_{i=1}^{N} \left[ (w^T x_i - w^{*T} x_i)^2 + \lambda \|w - w^*\|^2 \right],$$

$$= \frac{1}{2} [\|Xw - Xw^*\|^2 + \lambda \|w - w^*\|^2].$$

*Let $\kappa(\cdot)$ denote the condition number of a matrix. Then,*

1. *$\kappa(\nabla_w^2 \mathcal{H}) < \kappa(\nabla_w^2 \mathcal{L})$. Hence, Sobolev training improves the conditioning of the optimization problem and accelerates the convergence of gradient descent for the linear model.*

2. *Let $\hat{w}_{L^2}$ and $\hat{w}_{H^1}$ be the optimal parameters that minimize $\mathcal{L}$ and $\mathcal{H}$, respectively. Then $\mathbb{E}_{x \sim P_{data}} (\hat{w}_{H^1}^T x - w^{*T} x)^2 < \mathbb{E}_{x \sim P_{data}} (\hat{w}_{L^2}^T x - w^{*T} x)^2$, i.e., Sobolev training improves generalization error.*

*Proof.* One can easily compute the Hessians as:

$$\nabla_w^2 \mathcal{L} = X^T X,$$
$$\nabla_w^2 \mathcal{H} = X^T X + \lambda I.$$

Since the Hessians are symmetric, positive semidefinite, $\kappa(\nabla_w^2 \mathcal{L}) = \frac{\lambda_{\max}(\nabla_w^2 \mathcal{L})}{\lambda_{\min}(\nabla_w^2 \mathcal{L})}$ and $\kappa(\nabla_w^2 \mathcal{H}) = \frac{\lambda_{\max}(\nabla_w^2 \mathcal{L}) + \lambda}{\lambda_{\min}(\nabla_w^2 \mathcal{L}) + \lambda}$. Therefore, $\kappa(\nabla_w^2 \mathcal{H}) < \kappa(\nabla_w^2 \mathcal{L})$, indicating that Sobolev training leads to a faster convergence rate for gradient descent.

Let $\hat{w}_{L^2} = (X^T X)^{-1} X^T y$ and $\hat{w}_{H^1} = (X^T X + \lambda I)^{-1}(X^T y + \lambda w^*)$ be the optimal parameters. Then, both are unbiased estimator of $w^*$, i.e.,

$$\mathbb{E}(\hat{w}_{L^2}) = \mathbb{E}\left[ (X^T X)^{-1}(X^T y) \right] = (X^T X)^{-1} \mathbb{E}(X^T X w^* + X^T \epsilon)$$
$$= w^*,$$
$$\mathbb{E}(\hat{w}_{H^1}) = \mathbb{E}\left[ (X^T X + \lambda I)^{-1}(X^T y + \lambda w^*) \right]$$
$$= (X^T X + \lambda I)^{-1} \mathbb{E}(X^T X w^* + \lambda w^*)$$
$$= w^*.$$

The variance of each model can be computed as:

$$Var_{L^2} = \mathbb{E} \sum_i (\hat{w}_{L^2}^T x_i - w^{*T} x_i)^2 = \mathbb{E}\left[ (\hat{w}_{L^2} - w^*)^T X^T X (\hat{w}_{L^2} - w^*) \right]$$
$$= \mathbb{E}\left[ \epsilon^T X \{(X^T X)^{-1}\}^T (X^T X)(X^T X)^{-1} X^T \epsilon \right]$$
$$= \mathbb{E}\left[ tr(\epsilon^T X (X^T X)^{-1} X^T \epsilon) \right]$$
$$= \mathbb{E}\left[ tr(\epsilon \epsilon^T X (X^T X)^{-1} X^T) \right]$$
$$= tr(\mathbb{E}(\epsilon \epsilon^T) X (X^T X)^{-1} X^T)$$
$$= tr(X (X^T X)^{-1} X^T) = tr((X^T X) X^T X)$$
$$= tr(I)$$
$$= d,$$

and

$$Var_{H^1} = \mathbb{E} \sum_i (\hat{w}_{H^1}^T x_i - w^{*T} x_i)^2 = \mathbb{E} \left[ (\hat{w}_{H^1} - w^*)^T X^T X (\hat{w}_{L^2} - w^*) \right]$$

$$= \mathbb{E} \left[ \epsilon^T X \{(X^T X + \lambda I)^{-1}\}^T (X^T X)(X^T X + \lambda I)^{-1} X^T \epsilon \right]$$

$$= \mathbb{E} \left[ tr(\epsilon^T X \{(X^T X + \lambda I)^{-1}\}^T (X^T X)(X^T X + \lambda I)^{-1} X^T \epsilon) \right]$$

$$= \mathbb{E} \left[ tr(\epsilon \epsilon^T X \{(X^T X + \lambda I)^{-1}\}^T (X^T X)(X^T X + \lambda I)^{-1} X^T) \right]$$

$$= tr \left[ \mathbb{E}(\epsilon \epsilon^T) X \{(X^T X + \lambda I)^{-1}\}^T (X^T X)(X^T X + \lambda I)^{-1} X^T \right]$$

$$= tr \left[ X^T X \{(X^T X + \lambda I)^{-1}\}^T (X^T X)(X^T X + \lambda I)^{-1} \right]$$

$$= \sum_{i=1}^d \frac{\sigma_i^2}{(\sigma_i + \lambda)^2},$$

where $\sigma_i^2$ denote the eigenvalues of $X^T X$. Therefore, $Var_{H^1} < Var_{L^2}$. Since both biases are zero, by the standard bias-variance tradeoff argument,

$$\mathbb{E}_{x \sim P_{data}} (\hat{w}_{L^2}^T x - w^{*T} x)^2 = Var_{L^2} + \sigma^2,$$

$$\mathbb{E}_{x \sim P_{data}} (\hat{w}_{H^1}^T x - w^{*T} x)^2 = Var_{H^1} + \sigma^2,$$

and $\mathbb{E}_{x \sim P_{data}} (\hat{w}_{H^1}^T x - w^{*T} x)^2 < \mathbb{E}_{x \sim P_{data}} (\hat{w}_{L^2}^T x - w^{*T} x)^2$. $\qquad\square$

Although the proposition relies on the idealized assumption that the true parameter $w^*$ is known, it provides intuition for the Sobolev acceleration effect and the generalization ability, which we rigorously establish for the ReLU network.

# B  PROOF OF THEOREMS

**Theorem 2.6.** *[Comparison of the optimization landscapes] Assume 2.1, 2.2, 2.3 hold with $K = 1$. Let $\theta$ denotes the angle between $w$ and $w^*$. Then we have*

$$\nabla_w^2 \mathcal{L} = \frac{1}{2}I - \alpha(uu^\top - \cos(\theta)vu^\top + \sin^2(\theta)I)(I - vv^\top),$$

$$\nabla_w^2 \mathcal{H} = I - \alpha(2uu^\top - \cos(\theta)vu^\top + \sin^2(\theta)I)(I - vv^\top) \tag{2}$$

*where $\alpha = \frac{\|w^*\|}{2\pi\|w\|\sin(\theta)}$, $u = \frac{w^*}{\|w^*\|}$, and $v = \frac{w}{\|w\|}$. Furthermore, if $\frac{\|w^*\|\sin(\theta)}{\|w\|} < \frac{\pi}{2}$, then*

$$\kappa(\nabla_w^2 \mathcal{H}) = \frac{1}{1 - 4\alpha\sin^2(\theta)} < \frac{1}{1 - 3\alpha\sin^2(\theta)} = \kappa(\nabla_w^2 \mathcal{L}).$$

*Proof.* The population gradient of $\mathcal{L}$ is given in Tian (2017) by:

$$\nabla_w \mathcal{L} = \frac{1}{2}(w - w^*) + \frac{1}{2\pi}(\theta w^* - \frac{\|w^*\|}{\|w\|}\sin(\theta)w).$$

We computed the population gradient of $\mathcal{J}$ in the proof of Theorem 2.11 as:

$$\nabla_w \mathcal{J} = \frac{(\pi - \theta)}{2\pi}(w - w^*) + \frac{\theta}{2\pi}w.$$

Therefore, the Hessians of the loss functions can be written as:

$$\nabla_w^2 \mathcal{L} = \frac{1}{2}I - \alpha(uu^\top - \cos(\theta)vu^\top + \sin^2(\theta)I)(I - vv^\top),$$

$$\nabla_w^2 \mathcal{H} = \nabla_w^2(\mathcal{L} + \mathcal{J}) = I - \alpha(2uu^\top - \cos(\theta)vu^\top + \sin^2(\theta)I)(I - vv^\top),$$

where $\alpha = \frac{\|w^*\|}{2\pi\|W\|\sin(\theta)}$, $u = \frac{w^*}{\|w^*\|}$, and $v = \frac{w}{\|w\|}$.

Here, $\nabla_w^2 \mathcal{L}$ is a rank-2 perturbation of $(\frac{1}{2} - \sin^2(\theta))I$ and therefore has the eigenvalue $\frac{1}{2} - \sin^2(\theta)$ with multiplicity $n - 2$. Considering the perturbation lies in the subspace spanned by $u$ and $v$, we can easily see that $\nabla_w^2 \mathcal{L}v = \frac{1}{2}v$ and $\nabla_w^2 \mathcal{L}(u - \cos(\theta)v) = (\frac{1}{2} - 2\alpha\sin^2(\theta))(u - \cos(\theta)v)$. Thus, $\lambda_{\max}(\nabla_w^2 \mathcal{L}) = \frac{1}{2}$ and $\lambda_{\min}(\nabla_w^2 \mathcal{L}) = \frac{1}{2} - 2\alpha\sin^2(\theta)$.

Similarly, $\nabla_w^2 \mathcal{H}$ has the eigenvalue $1 - \alpha\sin^2(\theta)$ with multiplicity $n - 2$ and $\nabla_w^2 \mathcal{H}v = v$, $\nabla_w^2 \mathcal{H}(u - \frac{2}{3}\cos(\theta)v) = (1 - 3\alpha\sin^2(\theta))(u - \frac{2}{3}\cos(\theta)v)$ yields $\lambda_{\max}(\nabla_w^2 \mathcal{H}) = 1$, and $\lambda_{\min}(\nabla_w^2 \mathcal{H}) = 1 - 3\alpha\sin^2(\theta)$.

We obtain $\kappa(\nabla_w^2 \mathcal{L}) = \frac{1}{1 - 4\alpha\sin^2(\theta)}$ and $\kappa(\nabla_w^2 \mathcal{H}) = \frac{1}{1 - 3\alpha\sin^2(\theta)}$. Therefore, $\kappa(\mathcal{H}) < \kappa(\mathcal{L})$, if $\frac{\|w^*\|\sin(\theta)}{\|w\|} < \frac{\pi}{2}$. $\qquad\square$

**Corollary 2.9.** *[Single-GD-step improvement] Assume 2.1, 2.2, 2.3 hold with $K = 1$. Consider the two gradient descent updates*

$$w_{new}^{(1)} = w_{old} - \eta\nabla_w\mathcal{L}(w_{old}), \quad w_{new}^{(2)} = w_{old} - \eta\nabla_w\mathcal{H}(w_{old})$$

*with stepsize $\eta > 0$. Then there exists explicit functions $C = C(w_{old}, w^*) > 0$ and an explicit function $F = F(w_{old}, w^*; \eta)$ such that whenever $\eta \leq C$, $F > 0$ and*

$$\|w_{new}^{(2)} - w^*\| \leq \|w_{new}^{(1)} - w^*\| - F.$$

*Proof.* We compare

$$w_n^{(1)} = w_{n-1}^{(1)} - \alpha\nabla_w\mathcal{L},$$

$$w_n^{(2)} = w_{n-1}^{(2)} - \alpha\nabla_w(\mathcal{L} + \mathcal{J}).$$

The first gradient descent step yields:

$$\|w_n^{(1)} - w^*\|^2 = \|w_{n-1}^{(1)} - w^*\|^2 + \alpha^2\|\nabla_w\mathcal{L}\|^2 - 2\alpha\nabla_w\mathcal{L}^\top(w_{n-1}^{(1)} - w^*),$$

and we have

$$\alpha^2 \|\nabla_w \mathcal{L}\|^2 = \frac{\alpha^2}{4\pi^2} \begin{pmatrix} \|w^*\| \\ \|w\| \end{pmatrix}^\top \underbrace{\begin{pmatrix} (\theta - \pi)^2 + \sin(\theta)^2 - (\theta - \pi)\sin(2\theta) & \pi(\theta - \pi)\cos(\theta) - \pi\sin(\theta) \\ \pi(\theta - \pi)\cos(\theta) - \pi\sin(\theta) & \pi^2 \end{pmatrix}}_{N_1} \begin{pmatrix} \|w^*\| \\ \|w\| \end{pmatrix},$$

and

$$-2\alpha \nabla_w \mathcal{L}^\top (w - w^*) = -\frac{\alpha}{2\pi} \begin{pmatrix} \|w^*\| \\ \|w\| \end{pmatrix}^\top \underbrace{\begin{pmatrix} \sin(2\theta) + (2\pi - 2\theta) & (\theta - 2\pi)\cos(\theta) - \sin(\theta) \\ (\theta - 2\pi)\cos(\theta) - \sin(\theta) & 2\pi \end{pmatrix}}_{N_2} \begin{pmatrix} \|w^*\| \\ \|w\| \end{pmatrix}.$$

The second gradient descent step yields:

$$\|w_n^{(2)} - w^*\|^2 = \|w_{n-1}^{(2)} - w^*\|^2 + \alpha^2 \|\nabla_w \mathcal{L} + \nabla_w \mathcal{J}\|^2 - 2\alpha(\nabla_w \mathcal{L} + \nabla_w \mathcal{J})^\top (w_{n-1}^{(2)} - w^*),$$

and we have

$$\alpha^2 \|\nabla_w \mathcal{L} + \nabla_w \mathcal{J}\|^2$$
$$= \frac{\alpha^2}{4\pi^2} \begin{pmatrix} \|w^*\| \\ \|w\| \end{pmatrix}^\top \underbrace{\begin{pmatrix} 4(\theta - \pi)^2 + \sin(\theta)^2 - 2(\theta - \pi)\sin(2\theta) & 4\pi(\theta - \pi)\cos(\theta) - 2\pi\sin(\theta) \\ 4\pi(\theta - \pi)\cos(\theta) - 2\pi\sin(\theta) & 4\pi^2 \end{pmatrix}}_{N_3} \begin{pmatrix} \|w^*\| \\ \|w\| \end{pmatrix},$$

and

$$-2\alpha(\nabla_w \mathcal{L} + \nabla_w \mathcal{J})^\top (w - w^*)$$
$$= -\frac{\alpha}{2\pi} \begin{pmatrix} \|w^*\| \\ \|w\| \end{pmatrix}^\top \underbrace{\begin{pmatrix} \sin(2\theta) - 4(\theta - \pi) & (2\theta - 4\pi)\cos(\theta) - \sin(\theta) \\ (2\theta - 4\pi)\cos(\theta) - \sin(\theta) & 4\pi \end{pmatrix}}_{N_4} \begin{pmatrix} \|w^*\| \\ \|w\| \end{pmatrix}.$$

It is clear that $N_1, N_2, N_3,$ and $N_4$ are positive semidefinite.

Finally, we can obtain

$$\alpha^2 (\|\nabla L\|^2 - \|\nabla L + \nabla J\|^2)$$
$$= \frac{\alpha^2}{4\pi^2} \begin{pmatrix} \|w^*\| \\ \|w\| \end{pmatrix}^\top \underbrace{\begin{pmatrix} -3(\theta - \pi)^2 + (\theta - \pi)\sin(2\theta) & -3\pi(\theta - \pi)\cos(\theta) + \pi\sin(\theta) \\ -3\pi(\theta - \pi)\cos(\theta) + \pi\sin(\theta) & -3\pi^2 \end{pmatrix}}_{N_5} \begin{pmatrix} \|w^*\| \\ \|w\| \end{pmatrix}.$$

We can easily see that $N_5$ is negative semidefinite. Thus $\|\nabla L\|^2 \leq \|\nabla L + \nabla J\|^2$.

Since $\|\nabla_w \mathcal{L}\|^2 - \|\nabla_w \mathcal{L} + \nabla_w \mathcal{J}\|^2 \leq 0$, and $\nabla_w \mathcal{J}^\top (w_{n-1} - w^*) > 0$, we can easily see that

$$\alpha^2 (\|\nabla_w \mathcal{L}\|^2 - \|\nabla_w \mathcal{L} + \nabla_w \mathcal{J}\|^2) + 2\alpha(\nabla_w \mathcal{J}^\top (w_{n-1} - w^*)) > 0,$$

whenever $0 < \alpha < \frac{-2\nabla_w \mathcal{J}^\top (w_{n-1} - w^*))}{\|\nabla_w \mathcal{L}\|^2 - \|\nabla_w \mathcal{L} + \nabla_w \mathcal{J}\|^2}$. Indeed, the upper bound is a function of $\theta, \|w\|,$ and $\|w^*\|$. For instance, if $\theta = 0$, then $\|w\|^2, \|w^*\|^2$ are canceled out and the inequality is given by $0 < \alpha < \frac{4}{3}$.

$\square$

**Theorem 2.11.** *Assume 2.1, 2.2, 2.3 hold with $K = 1$. Consider the gradient flow $\dot{w} = -\nabla_w \mathcal{H}(w)$, where $\mathcal{H}$ is given in equation 2, and $V(w) = \|w - w^*\|_2^2$. Suppose that $\|w^0 - w^*\| < \|w^*\|$. Then,*

$$\frac{dV}{dt} = -(w - w^*)^\top \nabla_w \mathcal{H} \leq -(w - w^*)^\top \nabla_w \mathcal{L} - \lambda(\theta)(\|w\|^2 + \|w^*\|^2) < 0,$$

*where $\lambda(\theta) = (2\pi - \theta) - \sqrt{\theta^2 + (2\pi - \theta)^2 \cos^2 \theta} \geq 0$ with $\theta$ denoting the angle between $w$ and $w^*$. Therefore, the decay of $V$ is accelerated by using the Sobolev loss function. Moreover, since the rate of acceleration $\lambda(\theta)$ is an increasing function of $\theta \in [0, \pi/2)$, the strength of acceleration increases as $\theta$ increases.*

*Proof.* By definition, $\nabla_w \mathcal{H} = \nabla_w \mathcal{L} + \nabla_w \mathbb{E}(\frac{1}{2}\|\partial_x g(X;w) - \partial_x g(X;w^*)\|^2)$. We prove the theorem by computing an analytical formula for the gradient of the $H^1$ seminorm term. Note that $\partial_x g(x;w) = \partial\sigma(w^\top x)w$ contains the canonical subgradient $\mathbb{1}_{\{w^\top x > 0\}}w$. This gives

$$\nabla_w \mathcal{J} := \nabla_w \mathbb{E}\left(\frac{1}{2N}\|\partial_x g(X;w) - \nabla_x g(X;w^*)\|^2\right)$$

$$= \nabla_w \mathbb{E}\left(\frac{1}{2N}\sum_{j=1}^{N}\|\mathbb{1}_{\{w^\top x_j > 0\}}w - \mathbb{1}_{\{{w^*}^\top x_j > 0\}}w^*\|^2\right)$$

$$= \mathbb{E}\left(\frac{1}{N}\sum_{j=1}^{N}(\mathbb{1}_{\{w^\top x_j > 0\}}w - \mathbb{1}_{\{w^\top x_j > 0\}}\mathbb{1}_{\{{w^*}^\top x_j > 0\}}w^*)\right)$$

$$= \frac{1}{N}\sum_{j=1}^{N}\left(\mathbb{P}(w^\top x_j > 0)w - \mathbb{P}(\{w^\top x_j > 0\} \cap \{{w^*}^\top x_j > 0\})w^*\right)$$

$$= \frac{(\pi - \theta)}{2\pi}(w - w^*) + \frac{\theta}{2\pi}w,$$

where $\theta$ denotes the angle between $w$, and $w^*$.

Therefore,

$$\frac{dV}{dt} = -(w - w^*)^\top(\nabla_w(\mathcal{L} + \mathcal{J}))$$

$$= -(w - w^*)^\top\nabla_w\mathcal{L} - (w - w^*)^\top\left(\frac{(\pi - \theta)}{2\pi}(w - w^*) + \frac{\theta}{2\pi}w\right)$$

$$= -\begin{pmatrix}\|w^*\|\\\|w\|\end{pmatrix}^\top \underbrace{\begin{pmatrix} \sin(2\theta) + 2\pi - 2\theta & -(2\pi - \theta)\cos(\theta) - \sin(\theta) \\ -(2\pi - \theta)\cos(\theta) - \sin(\theta) & 2\pi \end{pmatrix}}_{=:M_1}\begin{pmatrix}\|w^*\|\\\|w\|\end{pmatrix}$$

$$\quad - \begin{pmatrix}\|w^*\|\\\|w\|\end{pmatrix}^\top \underbrace{\begin{pmatrix} 2\pi - 2\theta & -(2\pi - \theta)\cos(\theta) \\ -(2\pi - \theta)\cos(\theta) & 2\pi \end{pmatrix}}_{=:M_2}\begin{pmatrix}\|w^*\|\\\|w\|\end{pmatrix}$$

$$=: -\begin{pmatrix}\|w^*\|\\\|w\|\end{pmatrix}^\top (M_1 + M_2)\begin{pmatrix}\|w^*\|\\\|w\|\end{pmatrix}.$$

Simply computing the eigenvalues, we obtain that for $\theta \in [0, \pi/2)$, both $M_1, M_2$ are positive semidefinite. Especially for $M_2$,

$$0 = \det(M_2 - \lambda I)$$
$$= (2\pi - 2\theta - \lambda)(2\pi - \lambda) - (2\pi - \theta)^2 \cos^2\theta \tag{4}$$
$$= \lambda^2 - 2\lambda(2\pi - \theta) + 4\pi(\pi - \theta) - (2\pi - \theta)^2 \cos^2\theta. \tag{5}$$

So

$$\lambda = (2\pi - \theta) \pm \sqrt{(2\pi - \theta)^2 - 4\pi(\pi - \theta) + (2\pi - \theta)^2 \cos^2\theta} \tag{6}$$
$$= (2\pi - \theta) \pm \sqrt{\theta^2 + (2\pi - \theta)^2 \cos^2\theta}. \tag{7}$$

So

$$\lambda_{\min}(M_2) = (2\pi - \theta) - \sqrt{\theta^2 + (2\pi - \theta)^2 \cos^2\theta} \tag{8}$$
$$= \frac{(2\pi - \theta)^2(1 - \cos^2\theta) - \theta^2}{(2\pi - \theta) + \sqrt{\theta^2 + (2\pi - \theta)^2 \cos^2\theta}} \tag{9}$$
$$= \frac{((2\pi - \theta)\sin(\theta) + \theta)((2\pi - \theta)\sin(\theta) - \theta)}{(2\pi - \theta) + \sqrt{\theta^2 + (2\pi - \theta)^2 \cos^2\theta}}. \tag{10}$$

The numerator in the last expression is nonnegative since

$$(2\pi - \theta)\sin(\theta) - \theta \geq \frac{3\pi}{2}\sin(\theta) - \theta \geq 0 \quad \text{for all } 0 \leq \theta < \pi/2. \tag{11}$$

Hence $\lambda_{\min}(M_2) \geq 0$ for $0 \leq \theta < \pi/2$, and the conclusion follows. $\qquad\square$

**Theorem 2.13.** *Assume 2.1 and 2.3 hold. Suppose $g(x; w) = \big(\sigma(w^\top x)\big)^2$, a two-layer network with a single ReLU$^2$ node. Consider the gradient flow $\dot{w} = -\nabla_w \mathcal{I}(w)$, where $\mathcal{I}(w) = \mathcal{I}_1(w) + \mathcal{I}_2(w) + \mathcal{I}_3(w)$ is the $H^2$ population loss with $\mathcal{I}_1 = \mathcal{L}$, $\mathcal{I}_2 = \mathcal{J}$, and $\mathcal{I}_3 = \mathbb{E}\big(\|\nabla_x^2 g(x; w) - \nabla_x^2 g(x; w^*)\|^2\big)$. If $\|w^0 - w^*\| < \|w^*\|$ then,*

$$-(w - w^*)^\top \nabla_w \mathcal{I}_j(w) < 0, \text{ for } j = 1, 2, 3,$$

*and hence, the decay of $V = \|w - w^*\|^2$ is accelerated under the gradient flow minimizing the higher order Sobolev loss functions.*

*Proof.* We sequentially compute the analytical formulas of $\nabla_w \mathcal{I}_j(w)$.

$$\nabla_w \mathcal{I}_1(w) = \nabla_w \mathbb{E}\left(\frac{1}{2N}\sum_{j=1}^N (\sigma(w^\top x_j)^2 - \sigma(w^{*\top} x_j)^2)^2\right)$$

$$= \mathbb{E}\left(\frac{1}{N}\sum_{j=1}^N (\sigma(w^\top x_j)^2 - \sigma(w^{*\top} x_j)^2)\nabla_w(\sigma(w^\top x_j)^2)\right)$$

$$= \mathbb{E}\left(\frac{2}{N}\sum_{j=1}^N (\mathbb{1}_{w^\top x_j > 0}(w^\top x_j)^2 - \mathbb{1}_{w^{*\top} x_j > 0}(w^{*\top} x_j)^2)\mathbb{1}_{w^\top x_j > 0}(w^\top x_j)x_j\right)$$

$$= \mathbb{E}\left(\frac{2}{N}\sum_{\substack{w^\top x_j > 0}} (w^\top x_j)^2(w^\top x_j)x_j - \sum_{\substack{w^\top x_j > 0 \\ w^{*\top} x_j > 0}} (w^{*\top} x_j)^2(w^\top x_j)x_j\right)$$

Let $F(v, w) = \sum_{j=1}^N \mathbb{1}_{v^\top x_j > 0 \wedge w^\top x_j > 0}(v^\top x_j)(w^\top x_j)^2 x_j$, then $\nabla_w \mathcal{I}_1(w) = \frac{2}{N}\mathbb{E}(F(w, w) - F(w, w^*))$.

We consider an orthonormal basis $e = \frac{v}{\|v\|}$, $e_\perp = \frac{w/\|w\| - e\cos\theta}{\sin\theta}$, where $\theta = \angle(v, w)$, and any orthonormal set of vectors that span the rest. In this coordinate system, $e = (1, 0, \cdots 0)$, $v = \|v\|e$, $w = (\|w\|\cos\theta, \|w\|\sin\theta, 0, \cdots, 0)$, and any vector $x = (r\cos\phi, r\sin\phi, z_3, \cdots z_d)$, where $\phi = \angle(x, e)$, $r = \|x\|$. Then,

$$\mathbb{E}(F(v, w)) = N\int_{\mathbb{R}^{d-2}}\int_{-\frac{\pi}{2}+\theta}^{\frac{\pi}{2}}\int_0^\infty \|v\|r\cos\phi\|w\|^2 r^2\cos^2(\phi - \theta)\begin{pmatrix} r\cos\phi \\ r\sin\phi \\ z_3 \\ \vdots \\ z_d \end{pmatrix}\frac{e^{-r^2/2}}{2\pi}r\,dr\,d\phi\,dz_3\cdots dz_d$$

$$= \frac{N\|v\|\|w\|^2}{2\pi}\begin{pmatrix} \cos\theta(2\sin\theta + 2(\pi - \theta)\cos\theta) + (\pi - \theta) + \sin\theta\cos\theta \\ \sin\theta(2\sin\theta + 2(\pi - \theta)\cos\theta) \\ 0 \\ \vdots \\ 0 \end{pmatrix}$$

$$= \frac{N\|w\|^2}{2\pi}((\pi - \theta) + \sin\theta\cos\theta)v + \frac{N\|v\|\|w\|}{2\pi}(2\sin\theta + 2(\pi - \theta)\cos\theta)w$$

Thus, $\nabla_w \mathcal{I}_1(w) = \big(3\|w\|^2 w - \frac{\|w^*\|}{\pi}((\pi - \theta) + \sin\theta\cos\theta)w - \frac{\|w\|\|w^*\|}{\pi}(2\sin\theta + 2(\pi - \theta)\cos\theta)w^*\big)$. Now the first inequality follows. Let $G = (\pi - \theta) + \sin\theta\cos\theta$, $H = 2\sin\theta + 2(\pi - \theta)\cos\theta$, then

the first result follows :

$$-(w - w^*)^\top \nabla_w \mathcal{I}_1(w) = -\frac{1}{\pi}(3\pi(\|w\|^2 - \|w\|\|w^*\|)^2 + 2\|w\|\|w^*\|(3\pi\|w\|^2$$
$$+ 2\|w^*\|^2(G\cos\theta + H) - 2\|w\|\|w^*\|(3\pi + G + H\cos\theta)))$$
$$= -\frac{1}{\pi}(3\pi(\|w\|^2 - \|w\|\|w^*\|)^2) - \frac{1}{2\pi}\begin{pmatrix}\|w^*\| \\ \|w\|\end{pmatrix}^\top M \begin{pmatrix}\|w^*\| \\ \|w\|\end{pmatrix} < 0,$$

as

$$M = \begin{pmatrix} 2G\cos\theta + 2H & -(3\pi + G + H\cos\theta) \\ -(3\pi + G + H\cos\theta) & 6\pi \end{pmatrix}$$

is positive semidefinite ($M_{11} > 0, M_{22} > 0, det(M) \geq 0$) for $\theta \in [0, \pi/2)$.

Now, we consider $\nabla_w \mathcal{I}_2(w)$. Note that $\nabla_x g(x; w) = 2\mathbb{1}_{w^\top x > 0}(w^x)w$.

$$\nabla_w \mathcal{I}_2(w) = \nabla_w \mathbb{E}\left(\frac{1}{2N}\sum_{j=1}^N \|2\mathbb{1}_{w^\top x_j > 0}(w^\top x_j)w - 2\mathbb{1}_{w^{*\top} x_j > 0}(w^{*\top} x_j)w^*\|^2\right)$$

$$= \mathbb{E}\left(\frac{4}{N}\sum_{j=1}^N (\mathbb{1}_{w^\top x_j > 0}(w^\top x_j)(w^\top w)x_j + \mathbb{1}_{w^\top x_j > 0}(w^\top x_j)^2 w\right.$$

$$- \mathbb{1}_{w^\top x_j > 0 \wedge w^{*\top} x_j > 0}(w^{*\top} x_j)(w^\top w^*)x_j$$

$$\left. - \mathbb{1}_{w^\top x_j > 0 \wedge w^{*\top} x_j > 0}(w^\top x_j)(w^{*\top} x_j)w^*\right)$$

Let $F(v, w) = \sum_{j=1}^N \mathbb{1}_{v^\top x_j > 0 \wedge w^\top x_j > 0}(w^\top x_j)((v^\top w)x_j + (v^\top x_j)w)$, then $\nabla_w \mathcal{I}_2(w) = \frac{4}{N}\mathbb{E}(F(w, w) - F(w, w^*))$. We again consider the orthonormal basis containing $e = \frac{v}{\|v\|}, e_\perp = \frac{w/\|w\| - e\cos\theta}{\sin\theta}$. Then,

$$\mathbb{E}(F(v, w)) = N\int_{\mathbb{R}^{d-2}}\int_{-\frac{\pi}{2}+\theta}^{\frac{\pi}{2}}\int_0^\infty \|v\|\|w\|r\cos(\phi - \theta)(\|w\|\cos\theta x_j + r\cos\phi w)\frac{e^{-r^2/2}}{2\pi}\,r\,dr\,d\phi\,dz_3\cdots dz_d$$

$$= \frac{N\cos\theta\sin\theta}{2\pi}\|w\|^2 v + \frac{N\|v\|\|w\|}{2\pi}(\sin\theta + 2(\pi - \theta)\cos\theta)w$$

Thus, $\nabla_w \mathcal{I}_2(w) = 4\left(\|w\|^2 w - \frac{\cos\theta\sin\theta}{2\pi}\|w^*\|^2 w - \frac{\|w\|\|w^*\|}{2\pi}(\sin\theta + 2(\pi - \theta)\cos\theta)w^*\right)$. Let $G_1 = \sin\theta + 2(\pi - \theta)\cos\theta$. Consequently,

$$-(w - w^*)^\top \nabla_w \mathcal{I}_2(w) = -\frac{2}{\pi}\left(2\pi\|w\|^4 - \cos\theta\sin\theta\|w^*\|^2\|w\|^2 - G_1\|w\|\|w^*\|\cos\theta\right.$$

$$\left. - 2\pi\|w\|^3\|w^*\|\cos\theta + \cos^2\theta\sin\theta\|w^*\|^3\|w\| + \|w\|\|w^*\|^3 G_1\right)$$

$$= -\frac{2}{\pi}\left(2\pi(\|w\|^2 - \|w\|\|w^*\|\cos\theta)^2 + \|w\|\|w^*\|\frac{1}{2}\begin{pmatrix}\|w^*\| \\ \|w\|\end{pmatrix}^\top M_2 \begin{pmatrix}\|w^*\| \\ \|w\|\end{pmatrix}\right) < 0,$$

where

$$M_2 = \begin{pmatrix} 2G_1 + 2\cos^2\theta\sin\theta & -\cos\theta(G_1 + \sin\theta + 2\pi\cos\theta) \\ -\cos\theta(G_1 + \sin\theta + 2\pi\cos\theta) & 4\pi\cos\theta \end{pmatrix}$$

is positive semidefinite for $\theta \in [0, \pi/2)$.

Finally, we prove $-(w-w^*)^\top \nabla_w \mathcal{I}_3(w) < 0$. Note that $\nabla_x^2 g(x; w) = 2\mathbb{1}_{w^\top x > 0} ww^\top \in \mathbb{R}^{d \times d}$.

$$\nabla_w \mathcal{I}_3(w) = \nabla_w \mathbb{E}\left(\frac{1}{2N} \sum_{j=1}^N \|2\mathbb{1}_{w^\top x_j > 0} ww^\top - 2\mathbb{1}_{w^{*\top} x_j > 0} w^* w^{*\top}\|^2\right)$$

$$= \nabla_w \mathbb{E}\left(\frac{1}{2N} \sum_{j=1}^N trace((2\mathbb{1}_{w^\top x_j > 0} ww^\top - 2\mathbb{1}_{w^{*\top} x_j > 0} w^* w^{*\top})^\top (2\mathbb{1}_{w^\top x_j > 0} ww^\top - 2\mathbb{1}_{w^{*\top} x_j > 0} w^* w^{*\top})))\right)$$

$$= \nabla_w \mathbb{E}\left(\frac{2}{N} \sum_{j=1}^N (\mathbb{1}_{w^\top x_j > 0} \|w\|^4 - 2\mathbb{1}_{w^\top x_j > 0 \wedge w^{*\top} x_j > 0} (w^\top w^*)^2 + \mathbb{1}_{w^{*\top} x_j > 0} \|w^*\|^4)\right)$$

$$= \mathbb{E}\left(\frac{8}{N} \sum_{j=1}^N (\mathbb{1}_{w^\top x_j > 0} \|w\|^2 w - \mathbb{1}_{w^\top x_j > 0 \wedge w^{*\top} x_j > 0} (w^\top w^*) w^*)\right)$$

$$= \frac{8}{N} \sum_{j=1}^N \left(\mathbb{P}(w^\top x_j > 0) \|w\|^2 w - \mathbb{P}(w^\top x_j > 0 \wedge w^{*\top} x_j > 0)(w^\top w^*) w^*\right)$$

$$= 4\|w\|^2 w - \frac{4(\pi - \theta)}{\pi}(w^\top w^*) w^*,$$

where $\theta$ denotes the angle between $w$, and $w^*$. Hence,

$$-(w - w^*)^\top \nabla_w \mathcal{I}_3(w) = -\frac{4}{\pi}\big(\pi\|w\|^4 - (\pi - \theta)\|w\|^2\|w^*\|^2 \cos^2\theta$$

$$- \pi\|w\|^3\|w^*\| \cos\theta + (\pi - \theta)\|w\|\|w^*\|^3 \cos\theta)$$

$$= -\frac{4}{\pi}\big(\pi(\|w\|^2 - w^\top w^*)^2 + (w^\top w^*)\begin{pmatrix}\|w^*\| \\ \|w\|\end{pmatrix}^\top M \begin{pmatrix}\|w^*\| \\ \|w\|\end{pmatrix}\big) < 0,$$

where

$$M = \begin{pmatrix} 2(\pi - \theta) & (\theta - 2\pi)\cos\theta \\ (\theta - 2\pi)\cos\theta & 2\pi \end{pmatrix}$$

is positive semidefinite and $w^\top w^* > 0$ for $\theta \in [0, \pi/2)$. This completes the proof. $\qquad\square$

Next, we consider a bias-free two-layer ReLU network $h(x; W) = \sum_j^K \sigma(w_j^\top x)$. Following the assumptions made in Tian (2017), we assume the teacher parameters $\{w_j^*\}_{j=1}^K$ form a orthonormal basis and the student parameter $W$ is initialized as:

$$w_l = y w_1^* + \cdots + y w_{l-1}^* + x w_l^* + y w_{l+1}^* + \cdots + y w_K^* \tag{12}$$

for some $(x, y) \in \mathbb{R}^2$. Using the analytic formula for the gradients of the objective functions in equation 15, it is easy to see that the same parameterization is maintained along the gradient flows

$$\dot{w} = -\nabla_w \mathcal{L}(w) \quad \text{or} \quad \dot{w} = -\nabla_w \mathcal{H}(w). \tag{13}$$

Hence, the $w^*$-dependent initialization in equation 12 reduces the $N$-dimensional gradient flow to the following 2-dimensional flows in the reduced variables $(x, y)$:

$$L^2 \text{ gradient flow} : \begin{pmatrix}\dot{x} \\ \dot{y}\end{pmatrix} = -\mathbb{E}\begin{pmatrix}\nabla_x \mathcal{L} \\ \nabla_y \mathcal{L}\end{pmatrix},$$

$$H^1 \text{ gradient flow} : \begin{pmatrix}\dot{x} \\ \dot{y}\end{pmatrix} = -\mathbb{E}\begin{pmatrix}\nabla_x \mathcal{H} \\ \nabla_y \mathcal{H}\end{pmatrix}. \tag{14}$$

**Theorem 2.14.** *Assume 2.1, 2.2, 2.3 hold with $K \geq 1$. If the teacher parameters $\{w_j^*\}_{j=1}^K$ form an orthonormal basis and a student parameter $W$ is initialized to be*

$$w_l = y w_1^* + \cdots + y w_{l-1}^* + z w_l^* + y w_{l+1}^* + \cdots + y w_K^*, \tag{3}$$

*under the basis of $\{w_j^*\}_{j=1}^K$, where $(z, y) \in \Omega = \{z \in (0, 1], y \in [0, 1], z > y\}$, then the following holds.*

    (1) *The student parameter $w_l$ converges to $w_l^*$ under the gradient flow $\dot{w}_l = -\nabla_{w_l}\mathcal{H}(W)$, i.e., $(z, y)$ converges to (1,0).*

    (2) *Near $(z, y) = (1, 0)$, the gradient flows can be linearized to 2-d dynamical systems:*

$$L^2 \text{ gradient flow} : \begin{pmatrix} \dot{z} \\ \dot{y} \end{pmatrix}_{L^2} \approx -M_3 \begin{pmatrix} z - 1 \\ y \end{pmatrix},$$

$$H^1 \text{ gradient flow} : \begin{pmatrix} \dot{z} \\ \dot{y} \end{pmatrix}_{H^1} \approx -2M_3 \begin{pmatrix} z - 1 \\ y \end{pmatrix},$$

    *and the eigenvalues of $M_3$ are $\lambda_1(M_3) = \frac{\pi}{2}$, and $\lambda_2(M_3) = \frac{\pi}{2}(K + 1)$. Hence, $M_3$ is positive definite for all K, and the convergence is accelerated. (see Figure 1.)*

    (3) *When $z, y$ are initialized such that $z = y \in (0, 1]$, the $L^2$ gradient flow converges to the saddle point $z = y = z_{L^2}^* = \frac{1}{\pi K}(\sqrt{K - 1} - \arccos(\frac{1}{\sqrt{K}}) + \pi)$, and the $H^1$ gradient flow converges to the saddle point $z = y = z_{H^1}^* = \frac{1}{2\pi K}(\sqrt{K - 1} + 2\pi - 2\arccos(\frac{1}{\sqrt{K}}))$ and $z_{L^2}(t) = (z(0) - z_{L^2}^*)e^{-K/2t}$ and $z_{H^1}(t) = (z(0) - z_{H^1}^*)e^{-Kt}$. Hence, the convergence is accelerated.*

*Proof.* We begin the proof by computing the gradients of $L^2$ and $H^1$ loss functions with respect to $w$ are given by:

$$-\mathbb{E}(\nabla_{w_j}\mathcal{L}) = \frac{1}{2\pi}\sum_{j'=1}^{K}\left[(\pi - \theta_j^{j'*})w_{j'}^* + \|w_{j'}^*\|\sin(\theta_j^{j'*})\frac{w_j}{\|w_j\|}\right.$$
$$\left. - (\pi - \theta_j^{j'})w_{j'} - \|w_{j'}\|\sin(\theta_j^{j'})\frac{w_j}{\|w_j\|}\right],$$

$$-\mathbb{E}(\nabla_{w_j}\mathcal{H}) = \frac{1}{2\pi}\sum_{j'=1}^{K}\left[2(\pi - \theta_j^{j'*})w_{j'}^* + \|w_{j'}^*\|\sin(\theta_j^{j'*})\frac{w_j}{\|w_j\|}\right.$$
$$\left. - 2(\pi - \theta_j^{j'})w_{j'} - \|w_{j'}\|\sin(\theta_j^{j'})\frac{w_j}{\|w_j\|}\right],$$

(15)

where $\theta_j^{j'*} = \angle(w_j, w_{j'}^*)$, and $\theta_j^{j'} = \angle(w_j, w_{j'})$. By assumption, we can rewrite the parameters as $w_j = P_j w$, and $w_j^* = P_j w^*$ for some $w$, and $w^*$ and by the symmetry. Also, we can simply check that $\mathbb{E}(\nabla_{w_j}\mathcal{L}) = P_j\mathbb{E}(\nabla_w\mathcal{L})$ and $\mathbb{E}(\nabla_{w_j}\mathcal{H}) = P_j\mathbb{E}(\nabla_w\mathcal{H})$, so that the trajectory of $W$ keep the cyclic structure. Therefore, we only need to prove for one parameter, say $w_1 = zw_1^* + yw_2^* + \cdots + yw_K^*$. Under this setting, we can simplify the gradients in equation 15 by:

$$-2\pi\mathbb{E}\begin{pmatrix} \nabla_z\mathcal{L} \\ \nabla_y\mathcal{L} \end{pmatrix} = ((K - 1)(\alpha\sin(\phi^*) - \sin(\phi)) + \alpha\sin(\theta))\begin{pmatrix} z \\ y \end{pmatrix}$$
$$- \begin{pmatrix} -(\pi - \theta) + \pi z + (\pi - \phi)(K - 1)y \\ -(\pi - \phi^*) + \pi y + (\pi - \phi)(z + (K - 2)y) \end{pmatrix},$$

(16)

$$-2\pi\mathbb{E}\begin{pmatrix} \nabla_z\mathcal{H} \\ \nabla_y\mathcal{H} \end{pmatrix} = ((K - 1)(\alpha\sin(\phi^*) - \sin(\phi)) + \alpha\sin(\theta))\begin{pmatrix} z \\ y \end{pmatrix}$$
$$- 2\begin{pmatrix} -(\pi - \theta) + \pi z + (\pi - \phi)(K - 1)y \\ -(\pi - \phi^*) + \pi y + (\pi - \phi)(z + (K - 2)y) \end{pmatrix},$$

(17)

where $\theta \equiv \theta_j^{j*}$, $\phi \equiv \theta_j^{j'}$, $\phi^* \equiv \theta_j^{j'*}$ (for $j' \neq j$), and $\alpha \equiv \frac{1}{\|w_j\|} = \frac{1}{(z^2 + (k-1)y^2)^{1/2}}$.

Now, we state several useful lemmas from Tian (2017).

**Lemma B.1** (Lemma 3 in Tian (2017)). *Let $\theta = \theta_j^{j*}$, $\phi = \theta_j^{j'}$ and $\phi^* = \theta_j^{j'*}$. Then $\cos(\theta) = \alpha x$, $\cos(\phi^*) = \alpha y$, and $\cos(\phi) = \alpha^2(2xy + (K - 2)y^2)$ and the following holds in $\Omega_{\epsilon_0} = \{(x, y) : x \geq 0, y \geq 0, x \geq y + \epsilon_0\}$ for small $\epsilon_0$2:*

*(1)* $\phi, \phi^* \in [0, \pi/2]$, *and* $\theta \in [0, \theta_0)$, *where* $\theta_0 = \arccos(\frac{1}{\sqrt{(K)}})$.

*(2)* $\cos(\phi) = 1 - \alpha^2(x-y)^2$ *and* $\sin(\phi) = \alpha(x-y)\sqrt{(2 - \alpha^2(x-y)^2)}$

*(3)* $\phi^* \geq \phi$ *and equality holds only when* $y = 0$ *and* $\phi^* > \theta$.

*Proof.* The proof is provided in Tian (2017). □

**Lemma B.2.** *For the dynamics in equation 17, there exists $\epsilon_0 > 0$ so that the triangular region* $\Omega_{\epsilon_0} = \{(x, y) : x \geq 0, y \geq 0, x \geq y + \epsilon_0\}$ *is a convergent region. The flow goes inward for all three edges, and any trajectory starting in $\Omega_{\epsilon_0}$ stays in $\Omega_{\epsilon_0}$.*

*Proof.* The proof directly follows from the calculation in Tian (2017).

(1) If $y = 0, 0 \leq x \leq 1$, i.e., on the horizontal line, the $y$ dynamics is given by
$$f_1 \equiv -2\pi\mathbb{E}(\nabla_y\mathcal{H}) = -\pi(x - 1) \geq 0.$$

(2) If $x = 1, 0 \leq y \leq 1$, i.e., on the vertical line, the $x$ dynamics is given by
$$f_2 \equiv -2\pi\mathbb{E}(\nabla_x\mathcal{H})$$
$$= -(K - 1)(-\alpha\sin\phi^* + \sin\phi + 2(\pi - \phi)y) + \alpha\sin\theta - 2\theta < 0.$$

(3) If $x = y + \epsilon$, i.e., on the diagonal line,
$$f_3 \equiv \left\langle -2\pi\mathbb{E}\begin{pmatrix} -\nabla_x\mathcal{H} \\ \nabla_y\mathcal{H} \end{pmatrix}, \begin{pmatrix} 1 \\ -1 \end{pmatrix} \right\rangle$$
$$= 2\phi^* - 2\theta - 2\epsilon\phi + ((K - 1)(\alpha\sin\phi^* - \sin\phi) + \alpha\sin(\theta))\epsilon \geq 0.$$
□

**Lemma B.3.** *For the dynamics in equation 17, the only critical point (i.e., $\mathbb{E}\begin{pmatrix} \nabla_x\mathcal{H} \\ \nabla_y\mathcal{H} \end{pmatrix} = 0$) is* $(x, y) = (1, 0)$.

*Proof.* Suppose that $(x, y)$ is another critical point and let $\epsilon = x - y$. Note that $\epsilon - 1 + Ky > 0$ (See, Tian (2017)). Being a critical point, $f_3 = 0$ implies that
$$((K - 1)(\alpha\sin\phi^* - \sin\phi) + \alpha\sin\theta) = -\frac{2}{\epsilon}(\phi^* - \theta) + 2\phi.$$
Using $\phi \in [0, \pi/2], \phi^* \geq \phi, \phi^* > \theta$, we have
$$-2\pi\mathbb{E}(\nabla_y\mathcal{H}) = -2(\pi - \phi)(\epsilon - 1 + Ky) - 2(\phi^* - \phi) - \frac{2}{\epsilon}(\phi^* - \theta)y < 0,$$
hence a contradiction. □

Combining Lemmas 1-3, we conclude that $(z, y)$ converges to $(1, 0)$, by the Poincaré-Bendixson theorem.

For the second part of the theorem, we consider the linearized dynamics of equation 16 and equation 17 around the unique critical point $(z, y) = (1, 0)$. Let
$$f_4(z, y) = -2\pi\mathbb{E}(\nabla_z\mathcal{L}),$$
$$f_5(z, y) = -2\pi\mathbb{E}(\nabla_y\mathcal{L}).$$
Then, we obtain $f_4(1, 0) = 0, \frac{\partial f_4}{\partial z}(1, 0) = -\pi, \frac{\partial f_4}{\partial y}(1, 0) = -\frac{\pi}{2}(K - 1)$ and $f_5(1, 0) = 0, \frac{\partial f_5}{\partial z}(1, 0) = -\frac{\pi}{2}, \frac{\partial f_5}{\partial y}(1, 0) = -\frac{\pi}{2}K$. Combining these, the linearized equations are given as follows:
$$\begin{pmatrix} \dot{z} \\ \dot{y} \end{pmatrix}_{L^2} = -\mathbb{E}\begin{pmatrix} \nabla_z\mathcal{L} \\ \nabla_y\mathcal{L} \end{pmatrix} \approx -\begin{pmatrix} \frac{1}{2} & \frac{1}{4}(K - 1) \\ \frac{1}{4} & \frac{1}{4}K \end{pmatrix}\begin{pmatrix} z - 1 \\ y \end{pmatrix} \tag{18}$$
$$\begin{pmatrix} \dot{z} \\ \dot{y} \end{pmatrix}_{H^1} = -\mathbb{E}\begin{pmatrix} \nabla_z\mathcal{H} \\ \nabla_y\mathcal{H} \end{pmatrix} \approx -2\begin{pmatrix} \frac{1}{2} & \frac{1}{4}(K - 1) \\ \frac{1}{4} & \frac{1}{4}K \end{pmatrix}\begin{pmatrix} z - 1 \\ y \end{pmatrix}, \tag{19}$$

where the eigenvalues of the matrix $M_3 := \begin{pmatrix} \frac{1}{2} & \frac{1}{4}(K-1) \\ \frac{1}{4} & \frac{1}{4}K \end{pmatrix}$ are $\lambda_1 = \frac{1}{4}$, and $\lambda_2 = \frac{K+1}{4}$ and

$v_1 = \begin{pmatrix} k-1 \\ -1 \end{pmatrix}, v_2 = \begin{pmatrix} 1 \\ 1 \end{pmatrix}$ are the corresponding eigenvectors. Thus, $M_3$ is positive definite for all $K > 0$. Moreover, we can write the solution of the linearized equation as:

$$\begin{pmatrix} z(t) \\ y(t) \end{pmatrix}_{L^2} \approx c_1 v_1 e^{-\lambda_1 t} + c_2 v_2 e^{-\lambda_2 t},$$

$$\begin{pmatrix} z(t) \\ y(t) \end{pmatrix}_{H^1} \approx c_1 v_1 e^{-2\lambda_1 t} + c_2 v_2 e^{-2\lambda_2 t},$$

for some $c_1, c_2$, and hence, the convergence is accelerated.

For the last part of the theorem, we assume $z = y$, so that $\phi = 0, \theta = \phi^* = \arccos(\frac{1}{\sqrt{K}})$, and $\alpha = \frac{1}{\sqrt{K}z}$. Then,

$$-\mathbb{E}(\nabla_z \mathcal{L}) = -\frac{K}{2}(z - z_{L^2}^*), \tag{20}$$

$$-\mathbb{E}(\nabla_z \mathcal{H}) = -K(z - z_{H^1}^*), \tag{21}$$

where $z_{L^2}^* = \frac{1}{\pi K}(\sqrt{K-1} + \pi - \arccos(\frac{1}{\sqrt{K}}))$, as shown in Tian (2017), and $z_{H^1}^* = \frac{1}{2\pi K}(\sqrt{K-1} + 2\pi - 2\arccos(\frac{1}{\sqrt{K}}))$. Thus, we obtain $z_{L^2}(t) = (z(0) - z_{L^2}^*)e^{-K/2t}$ and $z_{H^1}(t) = (z(0) - z_{H^1}^*)e^{-Kt}$, and the conclusion follows. $\square$

**Theorem 2.15.** *If the teacher parameters $\{w_j^*\}_{j=1}^K$ form an orthonormal basis and the student parameters are initialized as the symmetric Toeplitz matrix:*

$$\begin{pmatrix} w_1 \\ w_2 \\ w_3 \\ \vdots \\ w_K \end{pmatrix} = \begin{pmatrix} t_1 & t_2 & t_3 & \cdots & t_K \\ t_2 & t_1 & t_2 & \cdots & t_{K-1} \\ t_3 & t_2 & t_1 & \cdots & t_{K-2} \\ \vdots & \vdots & \vdots & \ddots & \vdots \\ t_K & t_{K-1} & t_{K-2} & \cdots & t_1 \end{pmatrix} \begin{pmatrix} w_1^* \\ w_2^* \\ w_3^* \\ \vdots \\ w_K^* \end{pmatrix},$$

*then under the linearized $L^2$ and $H^1$ gradient flows, $\mathbb{T}_{L^2} = (t_1 - 1, t_2, \ldots, t_K)$ and $\mathbb{T}_{H^1} = (t_1 - 1, t_2, \ldots, t_K)$ follow*

$$\dot{\mathbb{T}}_{L^2} = -M\mathbb{T}_{L^2}, \quad \dot{\mathbb{T}}_{H^1} = -2M\mathbb{T}_{H^1}$$

*for a positive definite matrix $M$.*

*Proof.* The parameterization in equation 12 can be written as

$$[w_1, \ldots, w_K] = [w_1^*, \ldots, w_K^*] \begin{pmatrix} z & y & y & \cdots & y \\ y & z & y & \cdots & y \\ y & y & z & \cdots & y \\ \vdots & & & & \vdots \\ y & \cdots & y & y & z \end{pmatrix} \tag{22}$$

In general, this parameterization can be generalized by using a $K \times K$ symmetric Toeplitz matrix:

$$[w_1, \ldots, w_K] = [w_1^*, \ldots, w_K^*] \begin{pmatrix} t_1 & t_2 & t_3 & \cdots & t_K \\ t_2 & t_1 & t_2 & \cdots & t_{K-1} \\ t_3 & t_2 & t_1 & \cdots & t_{K-2} \\ \vdots & \vdots & \vdots & \ddots & \vdots \\ t_K & t_{K-1} & t_{K-2} & \cdots & t_1 \end{pmatrix}. \tag{23}$$

Hence, instead of $(x, y)$, each parameter $w_j$ is parameterized by $(t_1, \ldots, t_K)$. By the same argument using symmetry, it suffices to track one variable $w_1 = \sum_{i=1}^K t_i w_i^*$. It reduces to analyzing a $K$-dimensional ODE in the new variable $\mathbf{t} := (t_1, \ldots, t_{K-1})^\top$. Recall that

$$\dot{w}_1 = -\mathbb{E}\left[\nabla_{w_1} \mathcal{L}(w_1, \ldots, w_K)\right]. \tag{24}$$

Writing the Jacobian matrix $J = [\frac{\partial w_1}{\partial t_i}] \in \mathbb{R}^{d \times K}$, chain rule and the gradient flow of $w_1$ give

$$J\dot{\mathbf{t}} = -\mathbb{E}\left[\nabla_{w_1}\mathcal{L}(w)\right]. \tag{25}$$

If $J$ has full rank, then this induces the following ODE for $\mathbf{t}$:

$$\dot{\mathbf{t}} = -J^\dagger \mathbb{E}\left[\nabla_{w_1}\mathcal{L}(w)\right] = -\mathbb{E}[\nabla_{\mathbf{t}}\mathcal{L}(w)], \tag{26}$$

where $J^\dagger$ is the pseudo-inverse of $J$. Since $w_1^*, \ldots, w_K^*$ are orthonormal,

$$J = (w_1^* \quad \cdots \quad w_K^*) \in \mathbb{R}^{d \times K} \quad \text{and} \quad J^\dagger = (J^\top J)^{-1} J^\top = J^\top. \tag{27}$$

Thus, for each $1 \le j \le K$,

$$\dot{t}_j = \langle w_j^*, -\mathbb{E}[\nabla_{w_1}\mathcal{L}(w)]\rangle = -\mathbb{E}[\nabla_{t_j}\mathcal{L}(w)]. \tag{28}$$

From equation 15,

$$-\mathbb{E}(\nabla_{w_1}\mathcal{L}) = \frac{1}{2\pi}\sum_{j'=1}^{K}(\pi - \theta_j^{j'*})w_{j'}^* + \alpha \sin\theta_j^{j'*} w_1 - (\pi - \theta_j^{j'})w_{j'} - \sin(\theta_j^{j'})w_1, \tag{29}$$

$$-\mathbb{E}(\nabla_{w_1}\mathcal{H}) = \frac{1}{2\pi}\sum_{j'=1}^{K}2(\pi - \theta_j^{j'*})w_{j'}^* + \alpha \sin\theta_j^{j'*} w_1 - 2(\pi - \theta_j^{j'})w_{j'} - \sin(\theta_j^{j'})w_1, \tag{30}$$

where $\alpha = (\sum_j t_j^2)^{-1/2}$, $\cos\theta_1^{j'*} = \alpha t_{j'}$, $\cos\theta_1^{j'} = \alpha^2 \sum_{l=1}^{K} t_l t_{l+j-1}$.

Then, we can now compute the dynamics of $t_1$, i.e., $\dot{t}_1 = -\nabla_{t_1}\mathcal{L}$ and $\dot{t}_1 = -\nabla_{t_1}\mathcal{H}$ by:

$$-\mathbb{E}(\nabla_{t_1}\mathcal{L}) = \frac{1}{2\pi}\left[(\pi - \theta_1^{1*}) + \alpha t_1 \sum_{j'=1}^{K}\sin\theta_1^{j'*} - \sum_{j'=1}^{K}(\pi - \theta_1^{j'})t_{j'} - t_1\sum_{j'=1}^{K}\sin\theta_1^{j'}\right], \tag{31}$$

$$-\mathbb{E}(\nabla_{t_1}\mathcal{H}) = \frac{1}{2\pi}\left[2(\pi - \theta_1^{1*}) + \alpha t_1 \sum_{j'=1}^{K}\sin\theta_1^{j'*} - \sum_{j'=1}^{K}2(\pi - \theta_1^{j'})t_{j'} - t_1\sum_{j'=1}^{K}\sin\theta_1^{j'}\right], \tag{32}$$

and the dynamics for $t_j$, $j \ne 1$ are give by:

$$-\mathbb{E}(\nabla_{t_j}\mathcal{L}) = \frac{1}{2\pi}\left[(\pi - \theta_1^{j*}) + \alpha t_j \sum_{j'=1}^{K}\sin\theta_1^{j'*} - \sum_{j'=1}^{K}(\pi - \theta_1^{j'})t_{j'+j-1} - t_j\sum_{j'=1}^{K}\sin\theta_1^{j'}\right], \tag{33}$$

$$-\mathbb{E}(\nabla_{t_j}\mathcal{H}) = \frac{1}{2\pi}\left[2(\pi - \theta_1^{j*}) + \alpha t_j \sum_{j'=1}^{K}\sin\theta_1^{j'*} - \sum_{j'=1}^{K}2(\pi - \theta_1^{j'})t_{j'+j-1} - t_j\sum_{j'=1}^{K}\sin\theta_1^{j'}\right]. \tag{34}$$

Therefore, the linearized system around a critical point $(t_1, t_2, \ldots, t_K) = (1, 0, \ldots, 0)$ is given by:

$$\begin{pmatrix} \dot{t}_1 \\ \dot{t}_2 \\ \vdots \\ \dot{t}_{K-1} \\ \dot{t}_K \end{pmatrix}_{L^2} = -\mathbb{E}\begin{pmatrix} \nabla_{t_1}\mathcal{L} \\ \nabla_{t_2}\mathcal{L} \\ \vdots \\ \nabla_{t_{K-1}}\mathcal{L} \\ \nabla_{t_K}\mathcal{L} \end{pmatrix} \approx -\begin{pmatrix} 1/2 & 1/4 & \cdots & 1/4 \\ 1/4 & 1/2 & \cdots & 1/4 \\ \vdots & \vdots & \ddots & \vdots \\ 1/4 & \cdots & 1/2 & 1/4 \\ 1/4 & \cdots & 1/4 & 1/2 \end{pmatrix}\begin{pmatrix} t_1 - 1 \\ t_2 \\ \vdots \\ t_{K-1} \\ t_K \end{pmatrix} \tag{35}$$

$$\begin{pmatrix} \dot{t}_1 \\ \dot{t}_2 \\ \vdots \\ \dot{t}_{K-1} \\ \dot{t}_K \end{pmatrix}_{H^1} = -\mathbb{E}\begin{pmatrix} \nabla_{t_1}\mathcal{L} \\ \nabla_{t_2}\mathcal{L} \\ \vdots \\ \nabla_{t_{K-1}}\mathcal{L} \\ \nabla_{t_K}\mathcal{L} \end{pmatrix} \approx -2\begin{pmatrix} 1/2 & 1/4 & \cdots & 1/4 \\ 1/4 & 1/2 & \cdots & 1/4 \\ \vdots & \vdots & \ddots & \vdots \\ 1/4 & \cdots & 1/2 & 1/4 \\ 1/4 & \cdots & 1/4 & 1/2 \end{pmatrix}\begin{pmatrix} t_1 - 1 \\ t_2 \\ \vdots \\ t_{K-1} \\ t_K \end{pmatrix}. \tag{36}$$

The eigenvalues of $\begin{pmatrix} 1/2 & 1/4 & \cdots & 1/4 \\ 1/4 & 1/2 & \cdots & 1/4 \\ \vdots & \vdots & \ddots & \vdots \\ 1/4 & \cdots & 1/2 & 1/4 \\ 1/4 & \cdots & 1/4 & 1/2 \end{pmatrix}$ are $\frac{K+1}{4}$ and $\frac{1}{4}$ of multiplicity $k-1$, and the conclusion follows. $\qquad\square$

## C ADDITIONAL EXPERIMENTS

### C.1 ANALYTICAL FORMULAS FOR THE POPULATION GRADIENTS

We verify the analytical formulas for the population gradients presented in Section 2. We randomly sampled $w^*$ from the standard normal distribution and added a uniform random vector $e$ s.t. $\|e\| \leq \|w^*\|$ to $w^*$ to obtain $w = w^* + e$ and $\|w - w^*\| \leq \|w^*\|$. We employed two neural networks

$$g_i(x; w) = (\sigma(w^\top x))^i, \text{ and } g_i(x; w^*) = (\sigma(w^{*\top} x))^i,$$

for $i = 1, 2$ with parameters $w$, and $w^*$, respectively. Subsequently, the error between the analytical formula (e.g., $\nabla_w \mathcal{J}, \nabla_w \mathcal{I}_j$) and the Monte-Carlo approximation of the population loss under spherical Gaussian distribution was computed by varying the input dimensions and the number of samples. Figure 7 shows the log–log plots for mean square errors between the analytical formulas and the Monte-Carlo approximations. For example, we compute $\frac{1}{2N} \sum_{i=1}^{N} \nabla_w \|\nabla_x g(x_j; w) - \nabla_x g(x_j; w^*)\|_2^2$ using automatic differentiation and computed its discrepancy to $\nabla_w \mathcal{J}$. In all cases, the error decreased linearly as the number of samples increased. Moreover, the errors were sufficiently small even in relatively high dimensions.

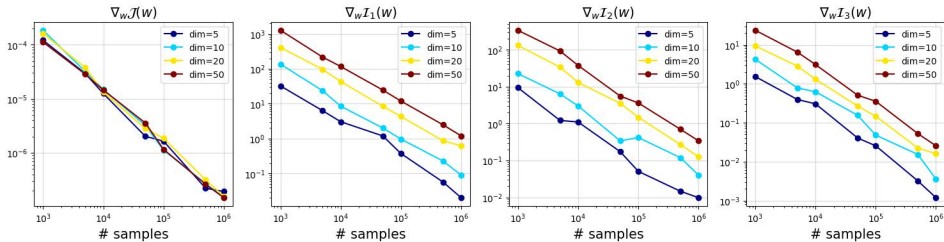

Figure 7: Log–log plots of the mean square errors(MSE) versus the number of samples. MSEs are computed between the analytical formulas $\nabla_w \mathcal{J}, \nabla_w \mathcal{I}_j$ and empirical expected values. Errors tend to decrease as the number of samples increases across all input dimensions.

### C.2 SOBOLEV TRAINING WITH APPROXIMATED DERIVATIVES

We provide comparative experiments on various numerical approximation schemes, including the Finite Difference Method (FDM) and Chebyshev spectral differentiation, to apply Sobolev training without derivative information. We consider a target function, the Acklev function $f(x, y) = -20 \exp(-0.2\sqrt{0.5(x^2 + y^2)}) - \exp(0.5(\cos(2\pi x) + \cos(2\pi y))) + e + 20, (x, y) \in [-2, 2]^2$, from Czarnecki et al. (2017).

We trained a neural network with 2-64-64-64-1 nodes and the hyperbolic tangent activation function using the Adam optimizer with a learning rate of $1e - 4$. The left panel of Figure 8 shows the errors during training in log scale for different loss functions: $L^2$, exact $H^1$, $H^1$ based on FDM, and $H^1$ based on the Chebyshev spectral differentiation. The proposed method achieved error levels almost equivalent to those in the exact derivative case, thereby suggesting that we can leverage the benefits of Sobolev training without requiring additional derivative information. Surprisingly, our method exhibited slightly faster convergence than the exact derivative case for this target function. In our experiments, the convergence speed of FDM is similar to that of

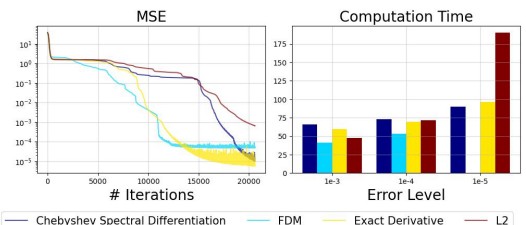

Figure 8: Left: Test MSEs for different loss functions during training. Right: Actual computation times to achieve certain error levels of [1e-3, 1e-4, 1e-5].

the exact $H^1$, but one can see that it cannot achieve the same error. The actual computation times required to achieve specific error levels of [1e-3, 1e-4, 1e-5] are presented in the right panel of Figure 8 for these loss functions. Owing to the efficient computation of tensor multiplication, our method

achieved error levels of [1e-3, 1e-4] without significantly increasing computation time compared to the $L^2$ loss function, and reached 1e-5 MSE considerably faster than the $L^2$ loss function.

### C.3 SOBOLEV TRAINING FOR AUTOENCODERS

We conduct experiments on a large-scale benchmark dataset, ImageNet, to train an autoencoder. In our setup, we resize the input data to (3, 64, 64) and use a batch size of 128. Both the encoder and decoder are implemented using the ResNet-18. The model is trained for 300 epochs, and the Adam optimizer is used with a learning rate of $1e - 3$.

The left panel of Figure 9 shows the convergence acceleration of the test error of the Sobolev training for the autoencoder task. We found that $H^1$ training performs significantly better than $L^2$ training in most iterations. The right panels show the original images and the reconstructed images of $L^2$ and $H^1$ training, respectively. In the first row, we found that the reconstructed image using $L^2$ training is blurry, whereas the reconstructed image using $H^1$ training is clearer. Furthermore, the second row shows that the $H^1$ training well reconstructs the image, whereas the $L^2$ training exhibits an unexpected background color.

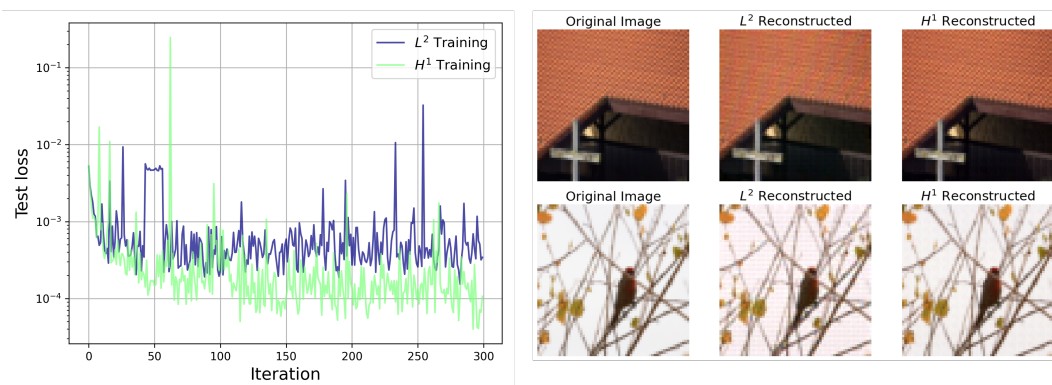

Figure 9: Results of the autoencoder task.

# D   COMPUTATIONAL COMPLEXITY OF $L^2$ AND $H^1$ LOSS FUNCTIONS

This section presents a detailed comparison of the computational complexity of the $L^2$ and $H^1$ loss functions in the diffusion model, including their theoretical time complexity and empirical measurements based on GFLOPs, peak memory usage, and loss computation time.

Let $u \in \mathbb{R}^{H \times W}$ denote the prediction error. The standard $L^2$ loss is defined as

$$\mathcal{L}_{L^2}(u) = \frac{1}{HW} \sum_{i=1}^{H} \sum_{j=1}^{W} u_{ij}^2, \tag{37}$$

which consists solely of element-wise operations. Thus, its computational complexity scales linearly with the number of pixels, i.e., $O(HW)$.

In contrast, the $H^1$ loss adopts a spectral formulation in which the error is evaluated in the Fourier domain. Let $\mathcal{F}_x$ and $\mathcal{F}_y$ denote the one-dimensional FFTs along the horizontal and vertical axes, respectively. The loss is computed as

$$\mathcal{L}_{H^1}(u) = \left\| \mathcal{F}_x^{-1} \left[ \mathcal{F}_x[u] \cdot (1 + \xi_x^2)^{1/2} \right] \right\|_2^2 + \left\| \mathcal{F}_y^{-1} \left[ \mathcal{F}_y[u] \cdot (1 + \xi_y^2)^{1/2} \right] \right\|_2^2, \tag{38}$$

where $\xi_x$ and $\xi_y$ denote the associated frequency multipliers. This computation requires applying FFTs of size $W$ for each of the $H$ rows and FFTs of size $H$ for each of the $W$ columns, resulting in an overall time complexity of

$$O\big(HW(\log H + \log W)\big).$$

The computational procedures for both losses, along with their per-stage time complexities, are summarized in Algorithm 1.

---

**Algorithm 1** $L^2$ and $H^1$ loss computation with their time complexity

---

**Require:** Prediction error $u \in \mathbb{R}^{H \times W}$

$L^2$ **loss**

$\quad \mathcal{L}_{L^2}(u) = \frac{1}{HW} \sum_{i,j} u_{ij}^2$ $\hfill \triangleright O(HW)$

$H^1$ **loss**

$\quad$ Compute frequency multipliers $\xi_x, \xi_y$ $\hfill \triangleright O(H + W)$

$\quad U_x \leftarrow \mathcal{F}_x[u]$ $\hfill \triangleright O(HW \log W)$

$\quad G_x \leftarrow U_x \cdot (1 + \xi_x^2)^{1/2}$ $\hfill \triangleright O(HW)$

$\quad g_x \leftarrow \mathcal{F}_x^{-1}[G_x]$ $\hfill \triangleright O(HW \log W)$

$\quad U_y \leftarrow \mathcal{F}_y[u]$ $\hfill \triangleright O(HW \log H)$

$\quad G_y \leftarrow U_y \cdot (1 + \xi_y^2)^{1/2}$ $\hfill \triangleright O(HW)$

$\quad g_y \leftarrow \mathcal{F}_y^{-1}[G_y]$ $\hfill \triangleright O(HW \log H)$

$\quad \mathcal{L}_{H^1}(u) = \|g_x\|_2^2 + \|g_y\|_2^2$ $\hfill \triangleright O(HW)$

---

To quantify the practical computational overhead of Sobolev training, we benchmarked both losses on a tensor of shape $(32, 3, 128, 128)$ using an NVIDIA RTX A6000 GPU, matching the setup in Section 3.4. Table 2 summarizes the GFLOPs, peak memory consumption, and loss computation times for the two losses.

Table 2: GFLOPs and peak GPU memory usage for $L^2$ and $H^1$ losses.

| Loss Function | GFLOPs | Peak Memory (MB) | Computation Time (s) |
|---|---|---|---|
| $L^2$ Loss | 0.003146 | 12.002 | 0.01234 |
| $H^1$ Loss | 0.220201 | 72.095 | 0.02871 |

