# OpenReview forum: "Sobolev acceleration for neural networks"
_ICLR.cc/2026/Conference — Submitted to ICLR 2026_

### Official Review · Reviewer_VVCF · 2025-10-21

**Soundness:** 3
**Presentation:** 2
**Contribution:** 2
**Rating:** 4
**Confidence:** 3

**Summary:**

This paper analyze shallow ReLU networks under a student-teacher framework with Gaussian inputs, deriving exact expressions for population gradients and Hessians. They prove that Sobolev training improves the condition number of the Hessian and accelerates gradient flow dynamics.  While this theoretical analysis offers valuable insights, its practical applicability is constrained by several strong assumptions.

**Strengths:**

This paper presents several key strengths, most notably its establishment of the first rigorous theoretical framework for Sobolev acceleration, a phenomenon previously supported only empirically. It offers considerable analytical depth by deriving exact formulas for population gradients and Hessians under a student-teacher setup. The work successfully identifies the improved conditioning of the Hessian as the core mechanism behind the acceleration, providing a clear and compelling explanation. Furthermore, the theoretical findings are substantiated by extensive and systematic experiments that demonstrate the phenomenon's persistence across various architectures, activation functions, and modern deep learning tasks like diffusion models.

**Weaknesses:**

The theoretical analysis relies on strong and idealized assumptions, including standard Gaussian inputs and shallow ReLU networks, which limits its direct applicability to real-world scenarios. The practical benefits are also contextualized with a limited baseline, as Sobolev training is compared only against standard L² loss, leaving its advantage over other advanced regularization or optimization techniques an open question. The argument for generalizability beyond the theoretical setting leans heavily on empirical results, as a formal extension to non-Gaussian data or deep architectures is not provided. Finally, while the reported computational overhead is low, a thorough theoretical analysis of the peak memory and computational cost associated with calculating the required derivatives is absent, which is crucial for assessing its practical efficiency.

**Questions:**

1. The theoretical results are derived under restrictive assumptions—standard Gaussian inputs, and shallow ReLU networks. It remains unclear whether the conclusions hold for more general data distributions or deeper ReLU networks.
2. While Sobolev training is compared to L^2 baselines, it would be valuable to include comparisons with other commonly used optimization methods,  regularization techniques or loss functions to better contextualize its benefits.
3. Some practical data do not follow a standard Gaussian distribution. The relationship between such data and the Gaussian assumption in the theory is not explained, raising questions about the practical relevance of the theoretical analysis.
4. The reported memory usage and per-epoch runtime for L^2 and H^1 training are remarkably close (Table 1). To better understand the computational overhead of Sobolev training, please provide a theoretical analysis and the peak training memory.

---

> ### Author Response · Authors · 2025-11-21
> **Response to Reviewer VVCF (1/2)**
>
> > **Strength : First rigorous theory for Sobolev acceleration, with clear Hessian-conditioning insight and broad experimental support.**
>
> **A.** Thank you for your thoughtful and detailed assessment of our work. We greatly appreciate that you accurately recognized the main contributions we intended, establishing a rigorous theoretical framework for Sobolev acceleration, identifying improved Hessian conditioning as its core mechanism, and supporting these findings with systematic experiments. Your careful reading and clear understanding of both the theoretical and empirical aspects are truly appreciated, and we are grateful for the time and effort you dedicated to this review. The corresponding revisions have been incorporated into the revised manuscript and are highlighted in blue.
>
> ---
>
> **Response to Weaknesses and Questions**
>
> ---
>
> > **W1 & Q1. The theory is limited by idealized assumptions, making its applicability to broader architectures and data distributions uncertain.**
>
> **A.** Thank you for raising this important point. We agree that the theoretical assumptions (Gaussian inputs and shallow ReLU networks) are restrictive. However, given the non-convex optimization nature of neural networks, such assumptions are necessary to obtain exact, closed-form expressions for population gradients and Hessians. Similar simplifying assumptions are standard in the literature; for example, widely used NTK analyses require infinite-width (or strong overparameterization) assumptions, and Gaussian input models are commonly adopted to make population integrals analytically tractable.
>
> Importantly, while our theory is developed under this analytically tractable regime, our experiments show that the same acceleration mechanism persists across much more general settings, including multilayer networks, CNN-based autoencoders, and diffusion models with non-Gaussian data. This strongly suggests that the phenomenon we identify reflects a structural property rather than an artifact of the assumptions.
>
> As our work provides the first theoretical step toward understanding Sobolev acceleration, we expect that future research will relax these assumptions and extend the analysis to increasingly general architectures and data distributions. We will discuss more about the data distribution in the response to Q3.
>
> > **W2 & Q2. The experiments compare Sobolev training only to $L^2$ loss, leaving its advantages over other optimization methods, regularization strategies, and loss functions unclear.**
>
> **A.** Thank you for the insightful comment.
> In our work, we made an effort to validate the robustness of Sobolev training across different optimization strategies.
> As shown in Figure 4, we employ Adam during the first half of the training epochs and L-BFGS during the remaining half, demonstrating that the effectiveness of Sobolev training is consistent regardless of the choice of optimizer.
> This supports the stability and general applicability of Sobolev training beyond a particular optimization method.
>
> Furthermore, following the experimental setup described in Section 3.2, we additionally compared Sobolev training against commonly used loss functions such as the Huber loss and the Charbonnier loss, and updated Figure 4 accordingly.
> For each loss function, we report both the test error and the training time, showing that the benefits of Sobolev training across different architecture.
> While $H^1$ training may begin with a slight delay compared to the other losses, it consistently converges to lower error levels and does so more rapidly as optimization proceeds.

---

> ### Author Response · Authors · 2025-11-21
> **Response to Reviewer VVCF (2/2)**
>
> > **W3 & Q3. The theory’s generalizability is uncertain because it assumes Gaussian inputs, yet many practical datasets are non-Gaussian and no formal extension beyond this setting is provided.**
>
> **A.** Thank you for pointing this out. The Gaussian assumption is used only to obtain closed-form expressions of population quantities such as $\mathbb{E}\left( w^T x \right)$, and $\mathbb{E}\left(xx^T \right)$, and similar terms appearing in the gradient and Hessian. Beyond enabling these integrals to be computed analytically, the proof does not rely on any special structural property of Gaussian data.
>
> In fact, the analysis can be extended to any input distribution for which these population moments can be computed in closed form (e.g., uniform or other analytically tractable distributions). The main limitation is that most practical data distributions do not yield such closed-form expectations, which makes exact theoretical analysis significantly more difficult. For this reason, we adopt the Gaussian model, which is the standard and most tractable choice in theoretical works on ReLU gradient dynamics.
>
> Importantly, the empirical results in our paper do not rely on Gaussian data: all experiments, including those on CNN-based denoising autoencoders and diffusion models, use non-Gaussian real-world datasets, yet the Sobolev acceleration phenomenon persists consistently. This strongly supports the practical relevance of the theoretical mechanism.
>
> We have added Remark 2.4 in the revised manuscript, clarifying that the Gaussian assumption is made purely for analytical tractability and that the framework extends in principle to any distribution for which the required moments can be computed.
>
> > **W4 & Q4. The paper reports low overhead but lacks a theoretical analysis of computational and peak-memory costs for Sobolev training.**
>
> **A.** We thank you for the valuable comment regarding the computational overhead of $H^1$ training compared to $L^2$ training.
> To better contextualize this overhead in practical diffusion model training, we incorporated your suggestion and expanded the timing analysis in Section 3.4, where we provide more detailed per-step measurements and reconstruct Table 1 accordingly.
> As summarized in Table 1 of the revised manuscript, the $H^1$ loss is approximately $2.3\times$ slower than the $L^2$ loss when measured alone.
> However, a single training step takes roughly one second, and the loss evaluation accounts for only 1-3% of that time.
> Given that each epoch contains 938 steps, the resulting difference in wall-clock training time is only a few seconds.
> Likewise, since the peak memory usage is dominated by the forward and backward passes of the diffusion model rather than by the loss computation itself, the choice between $L^2$ and $H^1$ losses has negligible impact on overall training memory requirements.
>
> In Appendix D, we provide a detailed theoretical analysis of both loss functions using Big-O notation and include Algorithm 1, which outlines their computational procedures and per-stage complexities.
> We also report additional measurements of GFLOPs, peak memory usage, and loss computation time, showing that the $H^1$ loss incurs greater cost due to its FFT-based spectral formulation, whereas the $L^2$ loss requires only element-wise operations.
> We thank you again for the insightful feedback, which helped us clarify these computational aspects more thoroughly.

---

> ### Comment · Reviewer_VVCF · 2025-11-25
>
> Thanks for your reply,
>
> 1.Can the theoretical results be extended to sub-Gaussian settings? We would also appreciate your insights on extending these findings to deeper ReLU networks.
>
> 2.It is essential to maintain consistent training configurations across all experiments to ensure reproducibility. A detailed description of the setup must be provided.
>
> 3. Please present the results of the cross-entropy loss function as a baseline.
>
> 4.As shown in Table 2, the peak memory usage of H¹ is over six times greater than that of L². Therefore, the claim that "the choice between H¹ and L² losses has a negligible impact on overall training memory" appears to be inconsistent with the data.

---

> ### Author Response · Authors · 2025-11-26
> **Response to reviewer VVCF**
>
> Thank you for your response and additional questions.
>
> > **Q1. Can the theoretical results be extended to sub-Gaussian settings? We would also appreciate your insights on extending these findings to deeper ReLU networks.**
>
> **A1.** Our results can be extended to general sub-Gaussian distributions. As noted in the revised manuscript (Remark 2.4), the Gaussian assumption was used only to obtain closed-form expectations such as $\mathbb{E}(ReLU(w^Tx))$. **The same analysis applies** as long as terms like $\mathbb{E}(ReLU(w^Tx))$ can be computed or approximated. Thus, the acceleration mechanism is not inherently tied to Gaussianity but to the ability to evaluate these population quantities.
>
> We also believe the mechanism should generalize, but analyzing deeper ReLU networks requires first understanding the gradient-flow behavior of $L^2$ training in multi-layer architectures. However, extending our analysis to networks with more hidden layers presents several significant challenges. The main difficulty is that, unlike the two-layer case, the weight vectors across different layers become strongly coupled: the gradient of one layer depends on the activations determined by all preceding layers. This inter-layer dependency makes it extremely difficult to obtain closed-form expressions for the
> population gradients and Hessians. Indeed, even for standard $L^2$ training, a full closed-form characterization of gradient dynamics for multi-layer ReLU networks is currently unavailable.
>
> We believe that a natural intermediate step is to first analyze multi-layer networks with a
> single neuron per layer, where the depth increases but the parameterization remains simple
> enough to keep the dynamics tractable. Once the $L^2$ population dynamics are understood in this setting, the extension to Sobolev training should follow in a similar manner to our current two-layer analysis.
>
> > **Q2. It is essential to maintain consistent training configurations across all experiments to ensure reproducibility. A detailed description of the setup must be provided.**
>
> **A2.** Thank you for emphasizing the importance of consistent training configurations for reproducibility. We fully agree. In our original paper, **all experimental configurations are explicitly documented**, including the GPU, neural network architectures, optimizers, learning rates, batch sizes, and other relevant hyperparameters.
>
> Moreover, **all accompanying code and scripts are available** in the supplementary materials, ensuring that every experiment can be reproduced exactly as reported. Nonetheless, we have carefully reviewed the manuscript to ensure that these details are clearly presented and easy to follow.
>
> > **Q3. Please present the results of the cross-entropy loss function as a baseline.**
>
> **A3.** Thank you for the helpful suggestion. We agree that cross-entropy loss is a widely used baseline in classification tasks. However, cross-entropy is fundamentally different from the $L^2$ objective we study.
>
> Unlike $L^2$ loss, cross-entropy does not directly measure the discrepancy between the network output and a target function. Instead, it corresponds to maximum-likelihood estimation under binomial or multinomial likelihoods, where the objective is to model class probabilities rather than to approximate a real-valued target function.
>
> Because of this structural difference, Sobolev training, which requires matching derivatives of the target function, cannot be naturally or meaningfully applied to cross-entropy objectives, as there is no well-defined target function whose gradients should be matched. **This makes cross-entropy an incompatible baseline for evaluating Sobolev acceleration** in our regression-focused framework.
>
> >**Q4. As shown in Table 2, the peak memory usage of $H^1$ is over six times greater than that of $L^2$. Therefore, the claim that "the choice between $H^1$ and $L^2$ losses has a negligible impact on overall training memory" appears to be inconsistent with the data.**
>
> **A4.** Thank you for raising this point. We believe **there was a misunderstanding regarding Table 2** in the revised manuscript. Although the ratio between the two peak memory measurements (72 MB vs. 12 MB) is indeed about six-fold, both values are still extremely small in absolute terms—only a few tens of megabytes.
>
> In the context of modern deep learning workloads, especially diffusion models, GPU memory usage typically ranges from several gigabytes to hundreds of gigabytes. Compared to these scales, an additional ~70 MB introduced by H¹ loss is negligible and does not meaningfully affect training feasibility, batch size, or hardware requirements.
>
> **Thank you again for your thoughtful questions and careful reading of our work. We believe we have thoroughly addressed all major concerns raised in your initial and second-round reviews. Should there be an opportunity for further clarification, we would be more than happy to provide it.**

---

### Official Review · Reviewer_gVmd · 2025-10-27

**Soundness:** 3
**Presentation:** 3
**Contribution:** 3
**Rating:** 6
**Confidence:** 3

**Summary:**

This work presents a theoretical framework proving that Sobolev training accelerates the convergence of Rectified Linear Unit (ReLU) networks. Under a student–teacher framework with Gaussian inputs and shallow architectures, they derive formulas for population gradients and Hessians, and quantify the improvements in conditioning of the loss landscape and gradient-flow convergence rates.

**Strengths:**

The paper is well written.

**Weaknesses:**

The assumptions (Assumption 2.2 (Two-layer ReLU network) and Assumption 2.3 (Gaussian population)) seem very restrictive.

**Questions:**

No question

---

> ### Author Response · Authors · 2025-11-21
> **Response to Reviewer gVmd**
>
> > **Stregth. The paper is well written.**
>
> **A.** Thank you for your positive evaluation, and we are grateful for the time and care you invested in reviewing our work. The corresponding revisions have been incorporated into the revised manuscript and are highlighted in blue.
>
> ---
>
> **Response to Weaknesses and Questions**
>
> ---
> > **W1. The assumptions (Assumption 2.2 (Two-layer ReLU network) and Assumption 2.3 (Gaussian population)) seem very restrictive.**
>
> **A.** Thank you for the positive evaluation and for raising this important point. We agree that the assumptions are restrictive. They are chosen to make the population gradients and Hessians exactly computable, which is essential for establishing the first rigorous characterization of Sobolev acceleration. Even for standard $L^2$ training, obtaining closed-form population derivatives beyond this setting is known to be extremely challenging.
>
> Importantly, while the theoretical framework focuses on this analytically tractable regime, we verified in our experiments that the same acceleration mechanism persists far beyond these assumptions: multi-neuron and multi-layer ReLU networks, CNN-based denoising autoencoders, and diffusion models all exhibit improved conditioning under Sobolev training. This suggests that the mechanism revealed by the simplified theory is structural rather than an artifact of the assumptions.
>
> To make this clearer, we have revised the manuscript emphasizing that the assumptions are used solely to enable exact analysis, while the empirical results demonstrate that the acceleration phenomenon remains robust in much more general architectures and data distributions.
>
> **Thank you again for your thoughtful questions and careful reading of our work. We believe we have thoroughly addressed all major concerns raised in your reviews. Should there be an opportunity for further clarification, we would be more than happy to provide it.**

---

### Official Review · Reviewer_SpAo · 2025-10-28

**Soundness:** 3
**Presentation:** 3
**Contribution:** 2
**Rating:** 4
**Confidence:** 3

**Summary:**

The paper’s main contribution is to provide theoretical guarantees on acceleration when using Sobolev training (Sobolev training adds first-order information to the loss function) for neural networks. The paper formally demonstrates this for the very particular case of ReLU networks with one neuron or multiple neurons in a one hidden layer network – in the case of multiple neurons, the results are on the gradient flow approximation. After the theoretical results, the paper shows multiple experimental results of Sobolev acceleration on settings well beyond the ones in the paper.

**Strengths:**

These are the strengths of the paper:
- The paper presentation of the new theoretical results is overall clear and easy to follow. There is enough description to understand what is going on in the paper.
- The overall writing of the paper is good and easy to understand.
- The paper grounds their assumptions on existing works in the past literature.

**Weaknesses:**

I have a series of comments, questions and concerns about the paper. I am willing to increase my score depending on the authors' response to them.


**>> Comments/concerns/questions:**
- The work (Cocola & Hand, 2020) is cited as previous work that has studied the convergence of Sobolev training. (Cocola & Hand, 2020) shows results that are probabilistic since they assume Gaussian random initialization of the weight parameters (**note that we are not talking about the data being Gaussian, but the weights being initialized as Gaussian**). This is the type of initialization that:
  1. is very common in recent theoretical studies on training convergence, such as (Du et al, 2018; Allen-Zu et al, 2019; Arora et al., 2019) – the three works mentioned in the introduction of the paper and which, in the case of the first two, consider networks that are not necessarily shallow.
  2. is **widely used in practice** (indeed, I am not aware of practical deep learning solutions that do not use random initialization, or even Gaussian initialization).
Then, what is the motivation for the authors to consider a different, deterministic, and arguably more restrictive type of initialization in their paper? Please note that (Cocola & Hand, 2020) already uses Gaussian initialization for Sobolev training, which is a more practical initialization
- Have the authors checked that none of the results by (Cocola & Hand, 2020) already indicate that Sobolev training could accelerate training dynamics? Can the authors discuss the possibility of using the results by (Cocola & Hand, 2020) in order to show Sobolev acceleration using Gaussian initialization of the weights?

- RELUs are not differentiable in their whole domain (they are not at the origin). How is it that one can simply write their gradient in equations like (1) without addressing this issue? How do the authors address the issue of the non-differentiability at the origin during their analysis?
- It is inconsistent to mention that $w$ is a matrix of dimension $d\times{K}$ in Assumption 2.2 while the statement of Theorem 2.5 refers to $w$ as a vector. Please, clarify.
- In Theorem 2.10, the function $\lambda(\theta)$ can potentially be zero, in which case, there will not be Sobolev acceleration. This seems to be for the specific value of $\theta = 0$, which means that both $w$ and $w^\*$ are parallel (zero angle, if I am not mistaken). It is possible that this could happen even when $w\neq w^\*$ since both vectors can have different magnitudes. Can the authors make a comment about this case, since this is the case where acceleration may not happen? The $||w^0-w^\*||<||w^\*||$ may help during initialization, but not necessarily during training; can the authors comment on that? Likewise, I am suspicious that something like this also happens in Theorem 2.11, though no specific function of $\theta$ is shown – is this true? Can the authors also make a comment on this?
- What are $\lambda_1$ and $\lambda_2$ in (2) from Theorem 2.12? They have not been defined before. How do they indicate positive definiteness of $M_3$?
- Theorem 2.13 is a linearized system, so **it is not** a general result for **global** convergence. Can the authors clarify this in the paper? If so, the dynamical system in Theorem 2.13 is only valid in a local neighborhood around $w^\*$ (which could be a very small one). This needs to be made explicit and addressed in the paper.
- The paper’s contribution and focus is theoretical, so I expect the experiments to point towards such results. I understand the authors have tried to do that to some extent in subsection 3.1., but I have an additional suggestion. If we look at the paper, all theoretical results for more than one neuron were done using gradient flows. Gradient flows, as I understood from the authors too, **do not** directly translate to gradient descent (GD) (and even less to stochastic gradient descent (SGD)). Thus, it would be interesting to simulate both the trajectories of the loss of the gradient flow, for which closed-form expressions are known as in (3) from Theorem 2.12., and compare it to the ones one would obtain by GD – both under the same initialization.
- Again, regarding the experiments, I understand the idea of experimenting settings beyond the one from the theoretical results; however, the experiments on autoencoders and diffusion models seem topicwise very remote from the rest of the paper – they feel out of place and do not seem to contribute to this theoretical paper as much. Moreover, I have two things to point out:
  - In the autoencoder experiments, it is mentioned that (Yu et al., 2023) has already studied them. How are the authors’ experiments different from (Yu et al., 2023)? Otherwise, what’s the contribution?
  - Has anyone ever used Sobolev training on diffusion models before the authors? If not, this is something new and a possible contribution. However, as I mentioned earlier, though being a contribution, it seems very far in scope from the theoretical part of the paper: diffusion models such as DDPM use U-Nets, which are vastly different in architecture than just feedforward ReLUs (like the ones in the theoretical part of the paper)! Indeed, a comprehensive empirical study of Sobolev training (and modifications of it) on different diffusion models could potentially result in its own paper that the authors could as well write and publish.



**>> Other things:**
- The first paragraph of the introduction should also mention the use of Neural Operators, which use neural networks for approximating a map between two functional spaces. They have been used in scientific computing applications–e.g., see Fourier Neural Operators and Deep Operator Networks (DeepONets).
- Lines 037-038: why is capturing the derivatives “an essential feature of many modern applications”? Citations are needed.
- Lines 066-067: it says that the condition number “governs the convergence rate of many optimization algorithms”. Which algorithms are those? References are needed.
- Lines 211-222: it is mentioned that the faster convergence for gradient flow “typically reflects a larger minimum eigenvalue of the Hessian”.A citation or a simple case illustrating this claim is needed.
- In Theorem 2.12, the symbol “$x$” is used as a coefficient. The problem is that this same symbol has been used before to denote the input data to the model. I suggest using a different symbol.
- The last part of Theorem 2.10 sounds a little bit informal and qualitative: it uses the expression “much more accelerated”. My suggestion is to remove the part after the last comma of the last sentence of Theorem 2.10, and simply say that “there is more acceleration as $\theta$ increases.”

**Questions:**

Please, see the Weaknesses section for most of my questions. This is one additional question:
- What sort of challenges and modifications the authors believe they can do to extend their analysis to networks with more hidden layers?

---

> ### Author Response · Authors · 2025-11-21
> **Response to Reviewer SpAo (1/5)**
>
> > **Strength. The paper is clearly written, easy to follow, and well-grounded in prior literature.**
>
> **A.** Thank you for your positive assessment of the clarity and organization of our paper. We sincerely appreciate your remarks that the presentation of the theoretical results is easy to follow, that the overall writing is clear, and that our assumptions are well-grounded in prior literature. Your careful reading and thoughtful evaluation are greatly appreciated. The corresponding revisions have been incorporated into the revised manuscript and are highlighted in blue.
>
> ---
>
> **Response to Weaknesses and Questions**
>
> ---
>
> >**W1. Why does the paper use a deterministic initialization instead of the more standard Gaussian initialization used in prior Sobolev-training work?**
>
> **A.** Thank you for this thoughtful question. We agree that Gaussian random initialization, as used in Cocola \& Hand (2020), is both standard in the NTK literature and widely adopted in practice. We also agree that such probabilistic initialization is natural for convergence analyses in the NTK regime.
>
> Our setting, however, differs fundamentally from the NTK framework. Cocola \& Hand operate in the infinite-width NTK regime, where the training dynamics linearize and the analysis focuses on the induced kernel. In contrast, our goal is to study finite-width networks, where the evolution of the weight vectors $w_1, w_2, \dots, w_K$ must be analyzed jointly. In this finite-dimensional and fully non-linear regime, the gradient flow dynamics are significantly more complex, and Gaussian initialization does not lead to a tractable dynamical system. To obtain exact population gradients and Hessians, and thereby provide the first rigorous characterization of Sobolev acceleration, we require a deterministic initialization that keeps the dynamics analytically manageable. In this sense, there is an inherent trade-off: NTK analyses use more practical initializations but rely on infinite width, whereas our analysis avoids infinite-width assumptions but must impose a more structured initialization.
>
> We fully acknowledge the reviewer’s point that extending the analysis to more general, practical random initializations is an important next step. We are actively exploring this direction. In particular, we believe that projecting the dynamics onto the teacher subspace, or analyzing convergence under small random perturbations around the deterministic initialization, may allow the arguments to be generalized to Gaussian or other commonly used initializations.
>
> > **W2. Do Cocola & Hand’s results already suggest Sobolev acceleration, and could their Gaussian-initialization theory be used to show it?**
>
> **A.** Thank you for raising a very important question. Yes, we have carefully examined Cocola \& Hand (2020). Their analysis provides important convergence guarantees for Sobolev training in the NTK regime, where Gaussian initialization is a natural choice. However, in their framework, the $L^2$ error dynamics under the Sobolev gradient flow cannot be isolated or written in a tractable closed form. As a consequence, establishing Sobolev acceleration (i.e., showing that the error $\Vert w - w^* \Vert$ decays faster under Sobolev training) is extremely difficult within their setting. In contrast, our student–teacher analysis yields explicit gradient and Hessian formulas, allowing us to track the evolution of the same error measure under both $L^2$ and Sobolev gradient flows. This enables a direct, rigorous comparison of convergence rates and makes it possible to formally establish Sobolev acceleration.
>
> > **W3. How do the authors justify using gradients of non-differentiable ReLUs, particularly at the origin, in their analysis?**
>
> **A.** ReLU non-differentiability at the origin does not cause issues in our analysis. All results in the paper are stated in terms of population gradients and Hessians, which involve expectations over a Gaussian distribution. Since a point drawn from $\mathcal{N}(0, I)$  hits the exact point $w^T x = 0$ with probability zero, the nondifferentiable set has measure zero and does not contribute to the integrals. Consequently, the population gradients and Hessians are well defined almost everywhere, and the resulting expressions coincide with the weak (distributional) derivatives typically used in the analysis of ReLU networks.
>
> > **W4. Clarify the inconsistency between treating $w$ as a $d \times K$ matrix and $\theta$ as a vector.**
>
> **A.** Thank you for pointing this out. We agree that the notation was inconsistent. We have revised Assumption 2.2 accordingly.

---

> ### Author Response · Authors · 2025-11-21
> **Response to Reviewer SpAo (2/5)**
>
> > **W5. Clarify cases where $\lambda(\theta)=0$ (or its analogue) prevents Sobolev acceleration and how such degenerate angle conditions may arise during training.**
>
> **A.** Thank you for pointing out this subtle issue. We agree that in Theorem 2.11 (Theorem 2.10 in the original manuscript), the function $\lambda(\theta)$, the minimum eigenvalue of $M_2$, satisfied $\lambda(0)=0$. However, this does not imply that the Sobolev acceleration term in $dV/dt$ vanishes when $\theta=0$. As shown in the proof, the contribution $M_2$ to $dV/dt$ is $-\frac{\left(\Vert w \Vert - \Vert w^* \Vert \right)^2}{2\pi} $ when $\theta =0 $, which is strictly negative whenever $w \neq w^*$. Therefore, even if two vectors are parallel, the acceleration never vanishes. In other words, $\lambda(\theta)$ represents a quantitative lower bound on the Sobolev acceleration effect. We have added Remark 2.12 to clarify this point more explicitly.
>
> >**W6. What are $\lambda_1$ and $\lambda_2$ in (2) from Theorem 2.12? They have not been defined before. How do they indicate positive definiteness of $H$?**
>
> **A.** Thank you for catching this omission. In Theorem 2.14 (Theorem 2.12 in the original manuscript), $\lambda_1$ and $\lambda_2$ refer to the eigenvalues of the matrix $M_3$. We will explicitly define them in the revised version. Since both eigenvalues are positive, $M_3$ is positive definite. We have revised the manuscript.
>
> > **W7. Clarify that Theorem 2.13 gives only local—not global—convergence due to its linearized dynamics.**
>
> **A.** Thank you for pointing this out. To make this explicit, we have added Remark 2.16 stating that the acceleration is a local effect.
>
> > **W8. Suggest comparing gradient-flow loss trajectories with GD trajectories under the same initialization.**
>
> **A.** We thank you for the constructive comments regarding the gradient flow analysis.
> To address this point more clearly, we provide two complementary visualizations.
>
> The first illustrates the vector fields corresponding to equations (18) and (19) in the original paper, enabling a direct comparison of the theoretical gradient flow under the $L^2$ and $H^1$.
> As shown in left and middle panels of Figure 1, one can observe that the vector fields induced by the $H^1$ metric exhibit noticeably larger magnitudes, indicating that points initialized near the target move toward it with greater speed.
> This demonstrates that trajectories governed by the $H^1$ gradient flow approach $(1,0)$ faster than those under the $L^2$ gradient flow.
>
> The second visualization presents empirical trajectories of $(z, y)$ obtained from our experiments, initialized according to the conditions specified in Theorem 2.14 (Theorem 2.12 in the original manuscript), where the $L^2$ and $H^1$ trajectories are compared under the same teacher parameters initialization.
> To empirically validate the gradient flow dynamics predicted in Theorem 2.14, we consider the $K=2$ and the $d=16$, with both NumPy and PyTorch random seeds fixed to zero for reproducibility.
> The parameters are initialized at $(z,y)=(0.6,0.1)$, consistent with the admissible initialization region in the theorem.
> We simulate the dynamics for $1000$ steps using a learning rate 0.01, and evaluate the gradients over a batch of $2000$ Gaussian input samples.
>
> As shown in right panel of Figure 1, the trajectory obtained under the $H^1$ training scheme converges to the target point $(1,0)$ significantly faster than the trajectory obtained under $L^2$ training, clearly illustrating the Sobolev acceleration.
>
> This observation aligns with your suggestion that, since the main contributions of the paper are theoretical and primarily formulated in terms of gradient flows rather than gradient descent it is important to provide evidence that mirrors the theoretical predictions.
> We sincerely thank you for this insightful comment, and the revised manuscript will incorporate the mentioned figure accordingly.

---

> ### Author Response · Authors · 2025-11-21
> **Response to Reviewer SpAo (3/5)**
>
> > **W9-1) Clarify how the autoencoder experiments differ from Yu et al. (2023) and what new contribution they provide.**
>
> **A.** We thank you for raising this point and for drawing our attention to the connection with Yu et al.\ (2023). Our autoencoder experiment is indeed inspired by their setup, but it is intentionally designed to be more challenging and more generalized than the one considered in Yu et al. Specifically, in Yu et al.\ (2023) the noise level at training and at test time is matched, whereas in our experiment we deliberately evaluate generalization under a mismatch in both noise amplitude and spatial frequency.
>
> Concretely, during training we corrupt the input images with $\epsilon_1 \sim \mathcal{N}(0, 1/4), \epsilon_2 = 0.3\sin\bigl(2\pi(x+y)\bigr)$,
> while at test time we use significantly stronger and more oscillatory noise, namely $\epsilon_1 \sim \mathcal{N}(0, 1), \epsilon_2 = 0.3\sin\bigl(20\pi(x+y)\bigr)$.
>
> Thus, the stochastic noise component has larger amplitude at test time (the standard deviation doubles from 0.5 to 1, and the deterministic component has the same amplitude 0.3 but much higher spatial frequency. This creates a substantially harder denoising task than the one seen during training, and we use this setup precisely to probe whether Sobolev training yields better out of distribution generalization than standard $L^2$ training.
>
> > **W9-2) Discuss whether Sobolev training has been used in diffusion models before and justify its relevance given their architectural distance from the theoretical setting.**
>
> **A.** We agree with you that the diffusion-model experiment is somewhat distant in scope from the theoretical part of the paper, which focuses on feedforward ReLU networks.
> Our intention in including this experiment is not to provide a comprehensive empirical study, but rather to offer a proof-of-concept demonstration of the feasibility of applying Sobolev training to more complex architectures such as U-Nets used in DDPM. In the revised version, we will clarify this positioning and emphasize that a systematic and in-depth investigation of Sobolev training (and its variants) for diffusion models is an interesting and important direction for future work, which we view as complementary to the theoretical contributions of the present paper.
>
> > **W10. Mention Neural Operators (FNOs, DeepONets) in the introduction as key methods for mapping between function spaces.**
>
> **A.** Thank you for the helpful suggestion. We have updated the first paragraph of the introduction.
>
> > **W11. Lines 037-038: why is capturing the derivatives “an essential feature of many modern applications”? Citations are needed.**
>
> **A.** Thank you for the suggestion. We have added appropriate citations to support the statement.

---

> ### Author Response · Authors · 2025-11-21
> **Response to Reviewer SpAo (4/5)**
>
> > **W12. Lines 066-067: it says that the condition number “governs the convergence rate of many optimization algorithms”. Which algorithms are those? References are needed.**
>
> **A.** Thank you for pushing us to make this connection explicit. While the global landscape under both $H^{2}$ and $L^{2}$-training are non-convex, our Theorem 2.6 identifies a specific basin of attraction (see Remark 2.8) where the Hessian of both losses are positive definite (so locally strongly convex) and the condition number for $H^{2}$ is strictly smaller than that for $L^{2}$ everywhere on that basin.
>
> Most popular first-order optimization algorithms converge linearly in a locally strongly convex landscape and the contraction rate depends critically on the condition number $\kappa$. A smaller $\kappa$ directly translates to fewer iterations required to reach a specific error tolerance. Below we list standard optimization algorithms where the convergence rate strictly depends on the condition number, clarifying why the reduced  $\kappa$ (as Sobolev training does) theoretically guarantees acceleration:
>
> **1. Gradient Descent (GD):**
> For $\mu$-strongly convex and $L$-smooth functions, the convergence rate of GD is linear and controlled by the condition number $\kappa = L/\mu$.
> The error at step $k$ satisfies $|w_k-w^\*|^2 \le \left(\frac{\kappa - 1}{\kappa + 1}\right)^k \|w_0 - w^\*\|^2$.
> As $\kappa$ decreases, the decay factor shrinks, exponentially reducing the iterations needed. (Ref: [1])
>
> **2. Conjugate Gradient (CG):**
> For quadratic problems (which local minima of neural networks resemble), CG converges significantly faster than GD, but its rate still depends on $\sqrt{\kappa}$ via the factor $\frac{\sqrt{\kappa} - 1}{\sqrt{\kappa} + 1}$. (Ref: [3])
>
> **3. Momentum and Nesterov's Accelerated Gradient (NAG):**
> Even accelerated methods are fundamentally limited by the condition number.
> The optimal rate for first-order methods is $O\\left((1 - \frac{1}{\sqrt{\kappa}})^k\right)$.
> While acceleration changes the dependency from $\kappa$ to $\sqrt{\kappa}$, a smaller $\kappa$ remains the primary driver of convergence speed. (Ref: [4,5])
>
> **4. Coordinate Descent (CD):**
> Algorithms that update parameters block-wise are sensitive to the maximum component-wise Lipschitz constant relative to strong convexity, effectively $\kappa_{CD} = \frac{\max_i L_i}{\mu}$. (Ref: [2,6])
>
> We have also revised the manuscript to include the references above (see below Remark 2.8).
>
> ---
>
> **References**
> 1. **Nesterov, Y. (2018).** *Lectures on Convex Optimization*. Springer.
>    (See Section 2.1.5 for GD and Section 2.2 for Accelerated Gradient.)
>
> 2. **Wright, S. J. (2015).** Coordinate descent algorithms. *Mathematical Programming*, 151(1), 3–34.
>
> 3. **Shewchuk, J. R. (1994).** *An Introduction to the Conjugate Gradient Method Without the Agonizing Pain.*
>    (See Section 7 for convergence bounds.)
>
> 4. **Polyak, B. T. (1964).** Some methods of speeding up the convergence of iteration methods.
>    *USSR Computational Mathematics and Mathematical Physics.*
>    (Introduced Heavy Ball momentum and its \\(\\kappa\\) dependency.)
>
> 5. **Beck, A. and Teboulle, M. (2009).**
>    A Fast Iterative Shrinkage-Thresholding Algorithm for Linear Inverse Problems.
>    *SIAM Journal on Imaging Sciences.*
>    (Establishes \\(1/\\sqrt{\\kappa}\\) rates for FISTA/Nesterov schemes.)
>
> 6. **Beck, A. and Tetruashvili, L. (2013).**
>    On the convergence of block coordinate descent type methods.
>    *SIAM Journal on Optimization*, 23(4), 2037–2060.

---

> ### Author Response · Authors · 2025-11-21
> **Response to Reviewer SpAo (5/5)**
>
> > **W13. Lines 211-222: it is mentioned that the faster convergence for gradient flow “typically reflects a larger minimum eigenvalue of the Hessian”. A citation or a simple case illustrating this claim is needed.**
>
> **A.** Thank you for this suggestion. We have revised the manuscript to clarify this relationship and added a citation to standard optimization literature (e.g., Boyd & Vandenberghe, 2004).To illustrate the claim simply: within the basin of attraction of a strict local minimum $w^\*$, the loss function can be approximated by a quadratic form:
> $$
> \mathcal{L}(w) \approx \mathcal{L}(w^\*) + \frac{1}{2}(w-w^\*)^\top H (w-w^\*)$$
> where $H = \nabla^2 \mathcal{L}(w^\*)$ is the Hessian. The gradient flow dynamics $\dot{w} = -\nabla \mathcal{L}(w)$ then become a linear differential equation:$$\dot{w} \approx -H(w-w^\*)$$The solution to this system is given by $w(t) - w^* = e^{-tH}(w(0) - w^\*)$. When decomposed into the eigenbasis of $H$, the error component along the $i$-th eigenvector decays as $e^{-\lambda_i t}$. Consequently, the overall asymptotic convergence rate is dominated by the slowest decay mode, which corresponds to the minimum eigenvalue $\lambda_{\min}$. Therefore, a larger $\lambda_{\min}$ mathematically guarantees a faster exponential decay rate for the gradient flow. This is explicitly visible in our Theorem 2.14, where the $L^2$ rate is governed by $e^{-\lambda t}$ and the Sobolev rate by $e^{-2\lambda t}$ (due to the doubled eigenvalues). We have also included references in the revised manuscript.
>
> ---
>
> **Reference**
>
> 1. **Boyd, S. & Vandenberghe, L. (2004).** Convex Optimization. Cambridge University Press. (Section 9.3, discussing the convergence analysis of gradient methods and the role of Hessian eigenvalues).
>
> ---
>
> > **W14. In Theorem 2.12, the symbol “$x$” is used as a coefficient. The problem is that this same symbol has been used before to denote the input data to the model. I suggest using a different symbol.**
>
> **A.** Thank you for pointing this out. To avoid confusion with the input variable, we have replaced the coefficient symbol $x$ with $z$ in the Theorem. The manuscript has been updated accordingly.
>
> >**W15. The last part of Theorem 2.10 sounds a little bit informal and qualitative: it uses the expression “much more accelerated”. My suggestion is to remove the part after the last comma of the last sentence of Theorem 2.10, and simply say that “there is more acceleration as $\theta$ increases.”**
>
> **A.** Thank you for the helpful suggestion. We have revised the statement accordingly and removed the informal phrasing.
>
> >**Q1. What sort of challenges and modifications the authors believe they can do to extend their analysis to networks with more hidden layers?**
>
> **A.** Thank you for the insightful question. Extending our analysis to networks with more hidden layers presents several significant challenges. The main difficulty is that, unlike the two-layer case, the weight vectors across different layers become strongly coupled: the gradient of one layer depends on the activations determined by all preceding layers. This inter-layer dependency makes it extremely difficult to obtain closed-form expressions for the population gradients and Hessians. Indeed, even for standard $L^2$ training, a full closed-form characterization of gradient dynamics for multi-layer ReLU networks is currently unavailable.
>
> We believe that a natural intermediate step is to first analyze multi-layer networks with a single neuron per layer, where the depth increases but the parameterization remains simple enough to keep the dynamics tractable. Once the $L^2$ population dynamics are understood in this setting, the extension to Sobolev training should follow in a similar manner to our current two-layer analysis.
>
>
> **Thank you again for your thoughtful questions and careful reading of our work. We believe we have thoroughly addressed all major concerns raised in your reviews. We also sincerely appreciate your positive overall feedback and your openness to reassessing the paper. If any further discussion becomes possible, we would be more than happy to engage and provide additional clarification.**

---

### Official Review · Reviewer_nnz1 · 2025-10-31

**Soundness:** 3
**Presentation:** 4
**Contribution:** 2
**Rating:** 4
**Confidence:** 3

**Summary:**

The paper intend to establish theoretical guarantees to the phenomenon known as "Sobolev Acceleration", which is the empirical observation that training neural networks with loss functions defined in Sobolev norms (i.e., including derivative of the target function) often leads to faster convergence compared to standard $L^2$-based training.

The authors focus on a one-hidden-layer ReLU neural network and assume that the input data follows a standard Gaussian distribution. The study begins with the simplest case of a ReLU network with a single ReLU neuron ($K=1$). The authors derive an exact analytical expression for the Hessian of the Sobolev loss function, then prove that the condition number of this Hessian is strictly smaller than that of the corresponding Hessian for the $L^2$ loss. This result implies that optimization algorithms such as gradient descent may converge more rapidly when trained with a Sobolev-type loss, due to the improved conditioning of the loss landscape.

The paper further study the dynamics of weight convergence, specifically on how the Sobolev loss influences the convergence rate of the squared parameter error $||w-w^\ast||$, where $w^*$ is assumed to be the true weight parameter of the ReLU network. It is shown that in the general case of more than one ReLU neuron ($K\geq 1$), Sobolev training will lead to a faster decay of the error $||w-w^\ast||$ (i.e., faster convergence of the weight parameter).
	​

**Strengths:**

1. To the best of my knowledge, this is the first paper that provides a theoretical verification of the Sobolev Acceleration phenomenon for neural networks from an optimization perspective. While the analysis focuses on a simplified setting (Gaussian input and a single hidden-layer ReLU network), it represents a meaningful step forward. Previous work (Lu et al., 2022) studied Sobolev acceleration only in the RKHS regime, rather than directly on neural networks.

2. Despite being primarily a theory paper, the authors include a comprehensive set of numerical experiments that convincingly support their theoretical claims. These experiments span multi-layer ReLU networks with different activation functions, CNN-based denoising autoencoders, and even diffusion models, which go well beyond the simple theoretical setup. Across all cases, the results consistently demonstrate that training with Sobolev loss leads to faster convergence than standard $L^2$-based training.

**Weaknesses:**

1. The paper's technical depth appears insufficient for ICLR. The theoretical analysis is restricted to a highly simplified setting, requiring the input distribution to be standard Gaussian and the neural network to have only one hidden layer. Moreover, the core result on Hessian conditioning is established only for network containing a single ReLU neuron, which limits its generality. Although the authors extend the weight convergence analysis to the multi-neuron case, this extension relies on the strong assumption that the true weight vectors $\{w_j^\ast\}$ form an orthonormal basis, which is rarely realistic in practical networks. Overall, these simplifications make the theoretical contributions feel narrow and somewhat lacking in depth relative to the expectations of ICLR-level theoretical work.

2. The paper clearly establishes that the Sobolev loss leads to a smaller Hessian condition number compared to the standard $L^2$ loss. However, the link between this improved conditioning and acceleration in convergence is not thoroughly explained. While the authors briefly mention that optimization algorithms such as gradient descent tend to converge faster when the loss landscape has a smaller condition number, no formal justification or citation is provided to support this claim. It would strengthen the paper to clarify whether this relationship is theoretically well established (and under what assumptions) or to provide references from the optimization literature. Moreover, it remains unclear whether the same argument extends to other optimization algorithms beyond vanilla gradient descent.

**Questions:**

I have an additional question:

In the manuscript, you mention that Czarnecki et al. (2017) study the benefits of Sobolev training from the perspective of sample complexity, showing that training with the Sobolev loss function can reduce the sample complexity of training. In contrast, your paper intends to verify the phenomenon of "Sobolev Acceleration" through optimization dynamics, focusing on the improved conditioning of the Hessian and the accelerated convergence of network parameters.
Could you comment on how these two perspectives (sample complexity vs optimization conditioning) relate to each other? Why is it important to study Sobolev training from the optimization point of view? Furthermore, do any of your theoretical results imply concrete consequences or bounds regarding the sample complexity of training under Sobolev losses?

---

> ### Author Response · Authors · 2025-11-21
> **Response to  Reviewer nnz1 (1/3)**
>
> > **Strength. This paper gives the first optimization-theoretic proof of Sobolev acceleration and provides broad empirical evidence—from ReLU nets to diffusion models—that Sobolev training consistently speeds up convergence.**
>
> **A.** Thank you for clearly identifying the strengths of our work and for the time and care you devoted to reviewing our manuscript. We are grateful that the intended contribution of the paper, providing theoretical results together with comprehensive experimental validation, was clearly conveyed, and we appreciate your thoughtful and thorough evaluation. The corresponding revisions have been incorporated into the revised manuscript and are highlighted in blue.
>
> ---
>
> **Response to Weaknesses and Questions**
>
> ---
>
> > **W1. The theoretical analysis is too simplified to meet ICLR’s expected technical depth.**
>
> **A.** We appreciate your thoughtful comments regarding the limitations of our theoretical setting. We fully agree that the current analysis is restricted to simplified assumptions, Gaussian inputs, a single-hidden-layer network, and the one-neuron Hessian characterization. These assumptions were chosen to make an exact, closed-form characterization of the population gradients and Hessian possible, which is extremely challenging in more general settings. To our knowledge, even for standard $L^2$ training, such exact formulas are not available beyond similarly simplified setups, and we made the first rigorous theoretical contribution to Sobolev acceleration. We thank you again for acknowledging the novelty and value of this contribution.
>
> Regarding the multi-neuron extension, we agree that the orthonormality assumption on the teacher weights is strong. Importantly, this assumption can be relaxed to the requirement that the weight vectors are merely linearly independent. In such cases, a Gram–Schmidt procedure yields an orthonormal basis spanning the same subspace, and the resulting coordinate transformation preserves the structure of the gradient and Hessian dynamics. Thank you for pointing out this issue.
>
> Although the theoretical framework is based on the simplest analytically tractable regime, the mechanism suggested by the theory, improved Hessian conditioning under Sobolev training, persists robustly across all of our experiments, including multi-neuron and multi-layer networks, CNN-based denoising autoencoders, and diffusion models. These empirical results show that the theoretical mechanism is not an artifact of the simplified assumptions, but rather a general phenomenon that extends well beyond the tractable setting. We also believe that this work provides the first comprehensive investigation (combining both rigorous theory and systematic experiments) of the Sobolev acceleration phenomenon, and we expect that it will serve as a foundation for future developments in this direction.

---

> ### Author Response · Authors · 2025-11-21
> **Response to  Reviewer nnz1 (2/3)**
>
> >  **W2. Unclear link between better conditioning and faster convergence.**
>
> **A.** Thank you for pushing us to make this connection explicit. While the global landscape under both $H^{2}$ and $L^{2}$-training are non-convex, our Theorem 2.6 identifies a specific basin of attraction (see Remark 2.8) where the Hessian of both losses are positive definite (so locally strongly convex) and the condition number for $H^{2}$ is strictly smaller than that for $L^{2}$ everywhere on that basin.
>
> Most popular first-order optimization algorithms converge linearly in a locally strongly convex landscape and the contraction rate depends critically on the condition number $\kappa$. A smaller $\kappa$ directly translates to fewer iterations required to reach a specific error tolerance. Below we list standard optimization algorithms where the convergence rate strictly depends on the condition number, clarifying why the reduced  $\kappa$ (as Sobolev training does) theoretically guarantees acceleration:
>
> **1. Gradient Descent (GD):**
> For $\mu$-strongly convex and $L$-smooth functions, the convergence rate of GD is linear and controlled by the condition number $\kappa = L/\mu$.
> The error at step $k$ satisfies $|w_k-w^\*|^2 \le \left(\frac{\kappa - 1}{\kappa + 1}\right)^k \|w_0 - w^\*\|^2$.
> As $\kappa$ decreases, the decay factor shrinks, exponentially reducing the iterations needed. (Ref: [1])
>
> **2. Conjugate Gradient (CG):**
> For quadratic problems (which local minima of neural networks resemble), CG converges significantly faster than GD, but its rate still depends on $\sqrt{\kappa}$ via the factor $\frac{\sqrt{\kappa} - 1}{\sqrt{\kappa} + 1}$. (Ref: [3])
>
> **3. Momentum and Nesterov's Accelerated Gradient (NAG):**
> Even accelerated methods are fundamentally limited by the condition number.
> The optimal rate for first-order methods is $O\\left((1 - \frac{1}{\sqrt{\kappa}})^k\right)$.
> While acceleration changes the dependency from $\kappa$ to $\sqrt{\kappa}$, a smaller $\kappa$ remains the primary driver of convergence speed. (Ref: [4,5])
>
> **4. Coordinate Descent (CD):**
> Algorithms that update parameters block-wise are sensitive to the maximum component-wise Lipschitz constant relative to strong convexity, effectively $\kappa_{CD} = \frac{\max_i L_i}{\mu}$. (Ref: [2,6])
>
> We have also revised the manuscript to include the references above (see below Remark 2.8).
>
> ---
>
> **References**
> 1. **Nesterov, Y. (2018).** *Lectures on Convex Optimization*. Springer.
>    (See Section 2.1.5 for GD and Section 2.2 for Accelerated Gradient.)
>
> 2. **Wright, S. J. (2015).** Coordinate descent algorithms. *Mathematical Programming*, 151(1), 3–34.
>
> 3. **Shewchuk, J. R. (1994).** *An Introduction to the Conjugate Gradient Method Without the Agonizing Pain.*
>    (See Section 7 for convergence bounds.)
>
> 4. **Polyak, B. T. (1964).** Some methods of speeding up the convergence of iteration methods.
>    *USSR Computational Mathematics and Mathematical Physics.*
>    (Introduced Heavy Ball momentum and its \\(\\kappa\\) dependency.)
>
> 5. **Beck, A. and Teboulle, M. (2009).**
>    A Fast Iterative Shrinkage-Thresholding Algorithm for Linear Inverse Problems.
>    *SIAM Journal on Imaging Sciences.*
>    (Establishes \\(1/\\sqrt{\\kappa}\\) rates for FISTA/Nesterov schemes.)
>
> 6. **Beck, A. and Tetruashvili, L. (2013).**
>    On the convergence of block coordinate descent type methods.
>    *SIAM Journal on Optimization*, 23(4), 2037–2060.

---

> ### Author Response · Authors · 2025-11-21
> **Response to  Reviewer nnz1 (3/3)**
>
> > **Q1. How do optimization benefits of Sobolev training relate to its sample-complexity effects?**
>
> **A.** Czarnecki et al. (2017) use the term sample complexity in a sense that is slightly different from the standard notion in statistical learning theory. Their focus is not on “how many samples are needed to learn a target function” but rather on “how many samples are needed to determine a target function”. To make this precise, they introduce the following definition: $$ K_{reg}(\mathcal{F}) := \min \{K: \forall f_w \in \mathcal{F}, ( f(x_i) = f_w(x_i) \forall i = 1, \dots, K ) \Rightarrow f=f_w \}$$ $$ K_{sob}(\mathcal{F}) := \min \{K: \forall f_w \in \mathcal{F}, ( f(x_i) = f_w(x_i) \wedge \nabla_x f(x_i) = \nabla_x f_w(x_i) \forall i = 1, \dots, K ) \Rightarrow f=f_w \}$$
> In other words, $K_{reg}(\mathcal{F})$ measures the minimum number of input locations that uniquely identify a target function $f$ within a function class $\mathcal{F}$. Under this definition, incorporating derivatives (as done in Sobolev training) increases the amount of information available at each input point, which effectively reduces the number of distinct samples needed to determine the target function.
>
> Our work, in contrast, studies Sobolev training from the optimization point of view. We show that once the target (function and derivatives) is fixed, incorporating derivative information fundamentally reshapes the optimization landscape by improving the conditioning of the Hessian and accelerating parameter convergence. Thus, whereas Czarnecki et al. address the amount of data needed to determine the target, our analysis concerns the efficiency with which gradient-based optimization approaches that target.
>
> Although these two viewpoints address different questions, they are complementary. Better conditioning leads to faster convergence toward the teacher parameters, which implies that for a fixed number of samples, Sobolev training can reach a low-risk region more rapidly than L2 training. This provides an optimization-based explanation for why models trained with Sobolev losses often achieve good generalization with fewer optimization steps, even when the number of training samples is fixed.
>
> In addition, in Appendix A of our manuscript, we analyze the linear setting and rigorously show that Sobolev training improves not only conditioning and convergence rate but also the expected generalization error. This result provides a direct bridge between the optimization and statistical perspectives: the improved conditioning induced by Sobolev regularization reduces the variance of the learned parameters, which can be interpreted as lowering the effective sample complexity required for good generalization.
>
> For general nonlinear ReLU networks, our present analysis focuses on optimization dynamics rather than explicit sample complexity bounds. Extending our results to derive formal statistical guarantees (e.g., uniform convergence or excess risk bounds for empirical Sobolev training) is an interesting direction for future work. Nonetheless, our findings indicate that the optimization mechanism we identify, improved Hessian conditioning, offers a complementary perspective that helps explain the empirical sample-efficiency benefits observed in prior studies.
>
> **Thank you again for your thoughtful questions and careful reading of our work. We believe we have thoroughly addressed all major concerns raised in your reviews. Should there be an opportunity for further clarification, we would be more than happy to provide it.**

---

### Author Response · Authors · 2025-11-22
**Authors’ Response to Reviewers’ Comments**

We would like to express our sincere gratitude to the reviewers for their careful evaluations and insightful comments. We have thoughtfully revised the manuscript in accordance with their valuable suggestions. We are especially thankful for the reviewers’ recognition of the main contributions of our work and for acknowledging the importance of establishing the first theoretical foundation for Sobolev acceleration.

Some reviewers noted the restrictive nature of our assumptions; however, these conditions were essential for deriving exact analytic expressions for the population gradients and Hessians, which enabled us to conduct the first rigorous analysis. We are also grateful that the reviewers encouraged us to clarify this point, and we emphasize that our extensive experiments demonstrate that the identified mechanism extends well beyond these assumptions. In our response, we further discussed how several assumptions, such as the orthogonality of weights and the Gaussian data distribution, can indeed be relaxed, helping to highlight the broader applicability of our analysis.

In addition, the revised manuscript includes several supplementary analyses and results that strengthen the overall contribution:

- **Computational cost analysis (Appendix D)**, providing a detailed comparison of the training-time overhead introduced by Sobolev acceleration.

- **Additional experiments comparing alternative loss functions**, including Huber and Charbonnier losses, presented in Figure 4 to contextualize the advantages of our method.

- **Expanded explanation of Figure 1**, clarifying the empirical behavior observed in the $H^1$ training dynamics.

- **Additional clarification in Remark 2.4, 2.12, 2.16**, elaborating on the interpretation of the theoretical results and addressing points raised by the reviewers.

All revisions in the manuscript are highlighted in blue.

---

### Author Response · Authors · 2025-11-30
**Authors' response to the Reviewers and the Area Chair**

We sincerely thank the reviewers and the area chair for their time and thoughtful evaluations. Below, we summarize **how we have addressed the major concerns** raised by each reviewer.

**1. Reviewer nnz1**
- We clarified why Gaussian inputs and one-hidden-layer ReLU networks are required for exact, closed-form population gradients/Hessians. We also explained that the orthonormality assumption can be relaxed to linear independence.
- Strengthened the link between improved conditioning and faster convergence; added citations and algorithm-specific explanations.
- We clarified the conceptual distinction between Czarnecki et al.’s “function determination” notion of sample complexity and our optimization-based analysis, and showed how improved conditioning complements and helps explain empirical sample-efficiency benefits.

**2. Reviewer SpAo**
- We clarified why deterministic initialization is required in our finite-width, fully nonlinear analysis (unlike NTK works relying on infinite width + Gaussian initialization), and why existing NTK-based results (e.g., Cocola & Hand, 2020) cannot establish Sobolev acceleration. We explicitly positioned our work as the first rigorous finite-width analysis enabling closed-form population gradients/Hessians.
- We explained that nondifferentiability at 0 is a measure-zero issue under Gaussian inputs and does not affect population derivatives.
- We clarified that acceleration does not vanish even when angles are small and added remarks explaining how Theorem 2.13 provides local (not global) convergence.
- Following the reviewer’s suggestion, we added visual comparisons between gradient-flow trajectories and GD trajectories under identical initialization, reinforcing the theoretical predictions.
- We clearly distinguished how our autoencoder experiments differ from Yu et al. (2023) (out-of-distribution generalization).
- We added all requested citations (Neural Operators, conditioning–convergence references, eigenvalue-based convergence explanation), and revised informal or ambiguous wording throughout.

**3. Reviewer gVmd**
- Clarified that the assumptions are used to make population gradients and Hessians exactly computable, enabling the first rigorous theoretical characterization of Sobolev acceleration.

**4. Reviewer VVCF**
- We clarified that Gaussian inputs and shallow ReLU networks are used solely to obtain exact, closed-form population gradients and Hessians, as is standard in the literature. We also added a remark explaining that the analysis extends in principle to any (including sub-Gaussian) distributions with computable moments.
- We expanded the experiments to compare Sobolev training with additional loss functions (Huber, Charbonnier) and reported both test errors and training times. We also emphasized that our results hold under multiple optimizers (Adam and L-BFGS), showing that the acceleration effect is robust to the choice of optimization method.
- We clarified that the Gaussian model is used purely for analytical tractability and not for the experiments. All empirical studies use non-Gaussian real-world datasets, yet still exhibit consistent Sobolev acceleration.
- We added a detailed complexity analysis (Appendix D) with Big-O formulas, an explicit algorithmic breakdown, and measurements of GFLOPs, loss computation time, and peak memory. We clarified that although the H¹ loss can be ~6× larger than L² in relative peak-memory terms (e.g., 72 MB vs. 12 MB) and about 2.3× slower in per-iteration computation time, these differences remain minor in the context of diffusion model training, as reflected in the revised Table 1.
- We confirmed that all experimental configurations (hardware, architectures, optimizers, hyperparameters) are fully documented and accompanied by code and scripts for exact reproducibility, and we improved the clarity of this documentation.

**We believe that all concerns raised by the reviewers, ranging from the assumptions and generality of the results to optimization-related justification, experiments, computational cost analysis, and connections to prior work, have been thoroughly addressed.  All corresponding revisions, clarifications, and references have been incorporated into the updated manuscript.**

**We also emphasize that this work provides one of the first rigorous theoretical foundations for Sobolev training and serves as an important starting point for further developments in this emerging research direction. We would be happy to provide any additional clarification if needed.**

---

### Meta-Review · Area_Chair_fPtY · 2025-12-28

**Summary:**

This paper establishes theoretical guarantees for the optimization dynamics of Sobolev training for two-layer neural networks with finite width. The theory can be used in an idealized setting to explain the Sobolev acceleration phenomenon observed across various network architectures.

The reviewers expressed constructive yet mixed strengths and weakness comments on this work. They pointed out the merit of being a novel rigorous theoretical work for explaining the phenomenon of Sobolev acceleration from the optimization perspective.

The reviewers also raised several common weakness and questions on the restrictive theoretical settings:
- setting for theoretical analysis is too restrictive: input Gaussian data, neural network with one hidden layer, ReLU activation, etc.;
- limited comparison with advanced regularization techniques other than the $L^2$ training;
- deterministic initialization v.s. the standard random Gaussian initialization;
- implications or connections to sample complexity for training the Sobolev loss in existing literature;
- the practical relevance of Sobolev training on diffusion models with current theoretical architecture analyzed in this paper.

**Reviewer Concerns:**

In response, the authors argued that their current analysis limitation to simplified assumptions, Gaussian inputs, a single-hidden-layer network, and one neuron Hessian characterization was to obtain an exact, closed form characterization of the population gradients and Hessian. These assumptions were made in parallel with the standard $L^2$ training.

Reading through the discussions, I appreciate that the rebuttal has thoroughly clarified their analysis limitation, partially addressed the reviewers’ concerns, and added additional experiments comparing alternative loss functions.

However, I still share the multiple concerns that the theoretical setting of this paper can be far from the real-world models, thus limiting the direct applicability of this work.

**Reviewer Scores:**

4 reviewers submitted their comments and scores (4/6/4/4) with confidence (3/3/3/3), with average score 4.5 and average confidence 3.

The positive score 6 has very brief justification, compared with other substantially more constructive comments. Even though some of the reviewers might have chance to raise their scores if a full discussion had continued, I am not 100% sure.

---

### Decision · Program_Chairs · 2026-01-26

Reject